# The 10 Most Crucial Circular Economy Challenge Patterns in Tourism and the Effects of COVID-19

Julia Martínez-Cabrera [1] and Francisco López-del-Pino [1,2,*]

[1] University Institute of Tourism and Sustainable Economic Development, University of Las Palmas de Gran Canaria, 35017 Las Palmas, Spain; julia.mcabrera95@gmail.com

[2] Department of Applied Economic Analysis, University of Las Palmas de Gran Canaria, 35017 Las Palmas, Spain

* Correspondence: francisco.lopez@ulpgc.es; Tel.: +34-928-45-81-16

**Abstract:** This paper makes a new contribution to the understanding of challenges for the transition toward the circular economy (CE) by identifying the main CE challenge patterns (CECPs) and analyzing their relevance for the tourism sector. Our work is based on a previous systematic literature review of 42 articles on CE through open coding following grounded theory. This allowed us to identify 68 CECPs and classify them into three levels of abstraction: microenvironmental, macroenvironmental, and organizational. To make this general research relevant to the tourism industry we conducted semi-structured interviews with 33 experts in CE and tourism, ensuring that theoretical saturation was reached. The data was analyzed in two coding phases, identifying which general CECPs are applicable to the tourism industry and which of them need further specification. The result shows that 34 of the 68 CECPs are applicable to tourism, of which 41% need to be specified to be relevant to the sector. Especially at the microenvironmental level, 53% of the general CECPs needed to be specified for the case of tourism. The analysis allowed to identify the 10 most crucial CECPs for the tourism industry and which of them have been most affected by the coronavirus (COVID-19) pandemic.

**Keywords:** circular economy; circular tourism; tourism; challenges; barriers; COVID-19



## 1. Introduction

By 2050, it is estimated that we will need three planets to provide the sufficient natural resources required by our current linear economic model [1]. Moving toward a circular economic model instead has the potential to "provide for every person's needs while safeguarding the living world on which we all depend" [2] (p. 45). Indeed, a circular economy (CE) is designed to systematically reduce the amount of resources required, reuse products to increase their lifespan, and regenerate natural systems.

Researchers argue that limited progress has been achieved in the transition toward a CE and attribute this to the variety of CE challenges [3–8]. Studying those challenges is perceived as an important lever for moving toward a CE model [3,9,10]. However, the nascent field on CE challenges is very fragmented and thus complex to leverage, with articles focusing on various aspects of CE challenges [11,12]. Therefore, they have called for further research to create a holistic perspective on the challenges hindering the transition toward a CE [6,11].

We have addressed this call of research in a previously published systematic literature review on CE challenges [13]. Thereby, the concept of patterns has played a major role in incorporating the different perspectives of the studies conducted around the various CE challenges. A pattern describes a problem that occurs over and over again in our environment [14]. Following this logic, we have defined circular economy challenge patterns (CECPs) as the core idea of recurring and similar CE challenges. We identified a total of

68 CECPs derived from our systematic literature review, studying 731 CE challenges mentioned within 42 articles across three different levels of abstraction: macroenvironmental, microenvironmental and organizational [13].

The CECPs represent a holistic framework to better comprehend the research performed in this field, thus helping academics and practitioners to understand the challenges of moving toward a CE.

Even though general research on CECPs is crucial to advance in this field, Kirchherr et al. [15] stress the importance of understanding how the general research for CE challenges can be made relevant for different industries. This is especially important as it enables industry-specific policymakers to take suitable interventions to accelerate the transition toward CE and to significantly promote wider adoption of the concept of CE [15]. Thus, we have decided to focus our research on the tourism industry for two reasons. On the one hand, the tourism industry is an important sector to study due to its economic importance and its negative environmental impacts. Tourism is considered the world's fastest-growing industry generating 10.3% of global GDP in 2018 [16], and plays a very important role in the economies of many countries around the world due to its multiplier effect [17]. However, the tourism industry is also a major contributor to environmental degradation due to its linear economic configuration, being responsible for 8% of global $CO_2$ emissions [18]. Subsectors such as transport (air travel, in particular, which is accountable for 40% of the emissions in tourism) and accommodation facilities are the ones that contribute most to global warming [19]. On the other hand, CE challenges in the tourism sector is a field that has been highly under-researched. During our work on the CECPs, we have only identified two articles studying CE challenges for the tourism industry [20,21]. However, these articles fall short in answering our research question in two main ways. First, Sørensen et al. [20] only mention five challenges that have been mapped to four CECPs, whereas Aryal [21] mentioned various challenges in a full text that were mapped to three CECPs. We argue that it is unlikely that saturation is reached, considering that we were able to identify up to 68 CECPs in total. Second, Sørensen et al. only interviewed 13 experts from Denmark and Aryal limited its scope to the tourism industry in Nepal, thus lacking the scale to derive generalizable challenges for the whole tourism industry. Therefore, we argue that further research on making the CECP literature relevant for the tourism industry is needed.

To tackle this research endeavor, several research questions arise: (i) Which are the CECPs that are applicable to the tourism industry? (ii) Which of those applicable CECPs need to be further specified, and how should they be specified for the tourism industry? (iii) Which are the most crucial CECPs to tackle for the tourism industry? (iv) Which negative effects does the COVID-19 pandemic have on the CECPs in tourism? This article represents an extensive study to answer these four research questions.

Following this introduction, this paper is structured as follows: Section 2 introduces the theoretical background by describing the concept of CE and circular tourism; Section 3 provides a summary of the previously conducted systematic literature review on CECPs; Section 4 describes the research methodology used; Section 5 outlines the findings on a macroenvironmental, microenvironmental, and organizational level as well as the COVID-19 effects on the CECPs; Section 6 presents a discussion around three key questions; finally, Section 7 provides a conclusion.

## 2. Theoretical Background

First, we outline the concept of circular economy and its origins. Second, we describe the novel concept of circular tourism.

### 2.1. Circular Economy

The concept of circular economy (CE) has received increasing attention over the last 10 years as an alternative to the current extractive take-make-dispose linear economy model [22]. The catalyst for this boost in popularity comes from the initial work in 2012

of the Ellen MacArthur Foundation [23], which has since then increased the amount of literature on CE by nearly 600% [24]. However, the CE is not as new as it might seem and cannot be traced back to one single author. The concept is rooted in industrial ecology and environmental/ecological economics [25], dating back to the 1960s with Boulding [26]. Boulding suggested the idea of improved durability by implementing a cyclical ecological system instead of a linear economic model to balance the economic activity with the earth's limited absorptive capacity. Nevertheless, according to many scholars, Pearce and Turner [27], inspired by Boulding's work, were the ones who first introduced the concept [25,28–30]. Later on came Benyus, who introduced another aspect and core building block of the CE: biomimicry [31]. Benyus proposed that the economic system can learn from and imitate (mimic) nature's ways to become more efficient and cope with the industrial and economic challenges [31]. Other schools of thought related to the CE concept are natural capitalism [32] and cradle to cradle [33]. The former seeks to create a shared economic platform that recognizes the needs of both the environment and the capital, whereas the latter considers all materials used in commercial and industrial processes as nutrients, of which there are two main categories: biological and technical [34].

When defining circular economy (CE), there are up to 114 different definitions that seek to explain this novel concept [35], although the most popular definition claimed by various authors [23,36] is that from the EMF, which states: "[CE] an industrial system that is restorative or regenerative by intention and design. It replaces the "end-of-life" concept with restoration, shifts towards the use of renewable energy, eliminates the use of toxic chemicals, which impair reuse, and aims for the elimination of waste through the superior design of materials, products, systems, and, within this, business models" [37] (p. 7).

### 2.2. Circular Tourism

The tourism industry is considered one of the most relevant contributors to GDP and employment worldwide [16]. Nevertheless, it is also the main source of environmental impacts, and many of the tourism externalities are related to high pressure on natural resources and increasing amounts of solid waste generation [38]. In this context, circular tourism could be seen as a way of approaching the study of the tourism sector, taking into account the principles of the circular economy. According to Girard and Nocca, a series of keywords such as "recovery, reuse, redevelopment, valorization and regeneration" [19] (p. 69) are linked to the concept of circular tourism, which they define as "a model able to create a virtuous circle producing goals and services without wasting the limited resources of the planet that are raw materials, water and energy" [19] (p. 68). Furthermore, different scholars outline that if the tourism industry wants to prosper within this new economic paradigm where nothing is waste, it is important that the whole tourism value chain adapts to this disruption by jointly collaborating with the different stakeholders of the industry and other industries [38,39].

Thus, it is not surprising that the concept of circular tourism has been slowly gaining momentum among scholars to support the industry in moving toward circularity. Rodríguez et al. [40] recently found out through an extensive literature review for the search period between 2009 and January 2020 that there are still only 55 articles and books published on the field of CE and tourism. Hence, many authors argue that state of the art in this field is in its infancy, with a reduced amount of literature available and a lack of shared understanding about CE and tourism [38,39,41–43].

### 3. Systematic Literature Review

In this section, we explain the previously conducted systematic literature review on CECPs as the foundation for our research [13]. The purpose of the systematic literature review was to identify the unspecific industry challenges impeding the transition toward a CE. For this, we followed the six review steps proposed by Paré et al. [44].

We reviewed 1106 papers within the databases of EBSCO-Business Source Complete, EBSCO-Academic Source Complete, and Web of Science Core Collection. After conducting

abstract screening and full-text screening, we selected a total of 42 papers for the body of research. We adopted an open-coding approach [45] to build the theory on existing industry unspecific challenges for CE. This allowed us to identify 731 CE challenges across the 42 selected papers. To cope with this large set of CE challenges and in order to create a holistic understanding of the CE challenges, we leveraged the concept of patterns. This concept has been leveraged in various fields of research, such as in software design [46], engineering [47,48], and business models [49].

Patterns require a certain level of generalization [50] and are used to describe the core idea [51] of recurring problems [52]. Based on this, we have defined CE challenge patterns (CECPs) as the core idea of recurring and similar CE challenges. This has allowed us to reduce the complexity and structure 68 CECPs within a larger framework.

The 68 CECPs were classified into three levels of abstraction and located into the elements of their corresponding frameworks:

(1) Macroenvironmental level refers to the elements that, from the company's perspective, cannot be changed. We used the well-known PESTEL framework [53,54] encompassed by the following elements and their abbreviations: political (P), economic (E), social (S), technological (T), environmental (EN), and legal (L).

(2) Microenvironmental level is defined as the factors that are external to the company but which belong to its industry. We differentiated between resources (R) that a company can purchase within its business environment (such as human, financial, physical, and intellectual resources on their respective markets), value chain (VC), defining the relationship between the different players within an industry, and infrastructure (I) that is established and can be leveraged by the players of the industry (such as roads, internet network, waste management systems, etc.).

(3) Organizational level refers to the factors inside the company. Here we leveraged three different frameworks that build upon each other to represent the different layers of abstraction from an organizational perspective. At the highest level, the ordering moments describe the coherent orientation and meaning of a company [55] and cluster the challenges related to the structure (STR), strategy (STRAT), and culture (CULT) of the company. At the middle level, the business model canvas [56] describes the core elements of the business logic of a company consisting of nine building blocks such as key partners (KP), key activities (KA), key resources (KR), value propositions (VP), customer relationship (CUSTR), customer segments (CUSTS), channels (CH), cost structure (CS), and revenue streams (RS). At the lowest level, the EMF systems diagram maps the looping actions derived from a circular business model and differentiates between the biological cycle (BC) or the technical cycle (TC).

Table 1 shows an overview of the 68 CECPs identified per level of abstraction and area.

**Table 1.** Overview of the 68 CECPs.

| Level | Area | CECP Names |
|---|---|---|
| Macroenvironmental | Political | P.01: (Political 01)—Missing adaptation and alignment of policies to local contexts |
| | | P.02: (Political 02)—Lack of international alignment and collaboration regarding policies and agreements |
| | | P.03: (Political 03)—Inefficient governmental structures |
| | | P.04: (Political 04)—Lack of adequate CE support by the government such as incentives/ funding, trainings and legislation |
| | | P.05: (Political 05)—Insufficient integration of CE in the political agenda and weak political commitment |
| | Economic | E.01: (Economical 01)—Tax system favors linear economy and does not support CE |
| | | E.02: (Economical 02)—Low economic development makes CE implementation at scale difficult |

**Table 1.** *Cont.*

| Level | Area | CECP Names |
|---|---|---|
| | Social | S.01: (Social 01)—Low level of awareness on the need for a more sustainable economy |
| | | S.02: (Social 02)—Society's aversion to change their current behavior, values and attitudes |
| | | S.03: (Social 03)—Missing enablement of the people towards CE |
| | | S.04: (Social 04)—Low acceptance to lack of ownership |
| | | S.05: (Social 05)—Conflicting interests with CE |
| | Technological | T.01: (Technological 01)—Existing technologies are not adapted to CE |
| | Environmental | EN.01: (Environmental 01)—Natural resources need costly and/or time-consuming treatment to be leveraged for CE |
| | | EN.02: (Environmental 02)—Geographical circumstances restrict applicability of CE solutions |
| | Legal | L.01: (Legal 01)—Lacking international and national legislation/regulation agreements and joined supportive frameworks for CE |
| | | L.02: (Legal 02)—Legislation not adapted to efficiently regulate CE |
| | | L.03: (Legal 03)—Insufficient implementation of CE regulations |
| Microenvironmental | Resources | R.01: (Resources 01)—Lack of experts on CE to hire and CE training offerings |
| | | R.02: (Resources 02)—CBMs have greater financial risks than linear business models |
| | | R.03: (Resources 03)—Cheaper price for virgin materials whereas environmentally friendly and looped materials are more expensive |
| | | R.04: (Resources 04)—Lack of efficient market to source available and high-quality CE resources |
| | | R.05: (Resources 05)—Lack of proof of solid CE theory, concepts, methods, measurements and role models (especially business models) |
| | | R.06: (Resources 06)—Accurate and reliable information is not available along SC making it difficult to implement looping actions |
| | | R.07: (Resources 07)—Information exchange regarding CE along the supply chain is not always possible |
| | Value Chain | VC.01: (Value Chain 01)—Misaligned profits, incentive systems and pressures along value chain making CE adoption unattractive |
| | | VC.02: (Value Chain 02)—Complex and costly to adapt the value chain to reverse logistics |
| | | VC.03: (Value Chain 03)—Lack of willingness and trust to collaborate across the value chain |
| | | VC.04: (Value Chain 04)—Dependency on other companies following a linear economy system hinders the transition towards CE of individual firms |
| | Infra-structure | I.01: (Infrastructure 01)—Information collection processes and information sharing platforms not in place/ not efficient enough |
| | | I.02: (Infrastructure 02)—Inefficient waste management/recycling systems, practices and infrastructures |
| | | I.03: (Infrastructure 03)—Difficult to implement CE spatial planning and transportation infrastructure |
| Organizational | Strategy | STRAT.01: (Strategy 01)—Shareholder interests not aligned with CE, lack of CE vision |
| | | STRAT.02: (Strategy 02)—Lack of transparency, forecast ability and difficult decision making |
| | Structure | STR.01: (Structure 01)—Missing organizational abilities and structure for CE |
| | | STR.02: (Structure 02)—Incentive systems misaligned with CE |
| | Culture | CULT.01: (Culture 01)—Lack of managerial commitment and leadership towards CE |
| | | CULT.02: (Culture 02)—Resistance to changes towards CE and conflict with the existing culture |

**Table 1.** *Cont.*

| Level | Area | CECP Names |
|---|---|---|
| | Customer segments | CUSTS.01: (Customer Segment 01)—Only niche markets for CE offerings |
| | | CUSTS.02: (Customer Segment 02)—Low willingness to pay for CE offerings |
| | | CUSTS.03: (Customer Segment 03)—Short-term fashion and technology trends |
| | Channels | CH.01: (Channels 01)—Challenge to establish and control CE compatible channels |
| | Customer Relationships | CUSTR.01: (Customer Relationships 01)—Difficulty in efficiently branding CE offerings and low status of looped offerings |
| | | CUSTR.02: (Customer Relationships 02)—Careless customer behaviour regarding offering without having ownership |
| | Value Proposition | VP.01: (Value Proposition 01)—Challenging to create efficient CE value proposition designs for complex offerings |
| | | VP.02: (Value Proposition 02)—CE offerings often inferior to linear economy offerings |
| | Key Partners | KP.01: (Key Partners 01)—Difficulties in finding suitable CE partners and building solid relationships with them |
| | | KP.02: (Key Partners 02)—High dependencies on partners represents a higher risk |
| | | KP.03: (Key Partners 03)—Lack of stakeholder cooperation |
| | Key Resources | KR.01: (Key Resources 01)—Lack of CE related technical resources and know-how |
| | | KR.02: (Key Resources 02)—Lack of financial resources to invest in CE |
| | | KR.03: (Key Resources 03)—Lack of CE skilled Human Resources within the company |
| | | KR.04: (Key Resources 04)—Lack of capacity to reinvent the company and its operations for a more CE like model |
| | | KR.05: (Key Resources 05)—Difficult to protect knowhow and intellectual property in CE models |
| | Key Activities | KA.01: (Key Activities 01)—Life Cycle Management of products is challenging |
| | | KA.02: (Key Activities 02)—Complex planning and forecast uncertainties |
| | Revenue Streams | RS.01: (Revenue Streams 01)—Cannibalization of sales due to new circular products |
| | | RS.02: (Revenue Streams 02)—Difficulties in establishing optimal pricing strategies |
| | Cost Structure | CS.01: (Cost Structure 01)—CE has higher operating costs |
| | | CS.02: (Cost Structure 02)—Financial and operational risk remains with the company instead of going to the customer |
| | | CS.03: (Cost Structure 03)—Very high upfront investment costs to implement CE |
| | Biological Cycles | BC.01: (Biological Cycles 01)—Bio-waste potential not fully exploited |
| | | BC.02: (Biological Cycles 02)—Challenge to safely return to biosphere |
| | Technical Cycles | TC.01: (Technical Cycles 01)—Infinite recyclability and material circularity is not always possible |
| | | TC.02: (Technical Cycles 02)—Ability to create high quality remanufacturing products |
| | | TC.03: (Technical Cycles 03)—Return flow uncertainty (quality, quantity, place, time...) |
| | | TC.04: (Technical Cycles 04)—Product complexity and design makes it difficult to recover value through recycling |
| | | TC.05: (Technical Cycles 05)—Looping actions might be less efficient than linear actions |

Source: [13].

While such a study on general CECPs is important to progress in this field, Kirchherr et al. have called for further studies to make the general research on CE challenges relevant for industries [15].

## 4. Methodology

The aim of this paper was to identify the main CECPs in the tourism industry. Due to the exploratory nature of this study, conducting interviews is considered a useful method for gaining a better understanding of the challenges that different stakeholders face when implementing CE in tourism [57]. To ensure qualitative rigor during the analysis of the data, we employed the systematic grounded theory approach [45]. The grounded theory involves the application of specific types of codes to the data through a parallel and iterative process of data collection and analysis, with the ultimate goal to generate theory [58].

### 4.1. Data Collection

For the data collection, we conducted semi-structured interviews with a total of 33 experts. We used a non-random judgment sampling approach to select the interviewees [15,59]. The sampling criteria were the following: (1) background—experts needed to be either knowledgeable in CE and active in tourism or knowledgeable in tourism and active in CE; (2) groups leading the CE transition—experts could belong to one of the groups categorized by Lieder and Rashid [60] and Bocken et al. [61] as the ones leading the transition toward a CE: policymakers, academic, businesses and organizations related to CE consultants, or be consultants due their broad expertise on a variety of topics across different sectors [62,63]; and (3) location—the sample should include experts from different countries in order to compensate the bias of having a majority of experts from Spain. Table 2 shows the selection of interviewees clustered by groups leading the CE transition and their geographic location.

**Table 2.** Overview of experts interviewed.

| Groups Leading the CE Transition | Spain | Rest of the EU | Outside the EU | Total |
|---|---|---|---|---|
| Policymakers | 4 | 3 | 2 | 9 |
| Academics | 4 | 2 | 1 | 7 |
| Consultants | 3 | 3 | 1 | 7 |
| Businesses | 3 | 2 | 1 | 6 |
| Organizations related to CE | 1 | 1 | 2 | 4 |
| Total | 15 | 11 | 7 | 33 |

Furthermore, Figure 1 shows the percentage of experts by groups leading the CE transition, that are on the one hand, knowledgeable in CE and active in tourism (a total of 51.5%) and on the other hand, that are knowledgeable in tourism and active in CE (a total 48.5%). This was based on: (1) their job title and job description in LinkedIn, which had to mention a specific role related to tourism or CE, to allow us to determine whether they were active in CE or in tourism and (2) during the interview, the second background information question showed in Table 3, "how are you related to the topic of circular economy and tourism?", allowed us to determine if the knowledge was predominantly in CE or in tourism. We observe in Figure 1 that most of the experts knowledgeable in CE and active in tourism belong to the group of "policymakers" (35%), whereas academics and businesses make 62% of the experts knowledgeable in tourism and active in CE.

Experts were searched within the social network LinkedIn and were reached directly by the leading researcher and by the experts' referencing (using the snowball sampling technique [64,65]) to expand our sample. A total of 96 CE experts were contacted, out of which 33 accepted to do the interview, resulting in a success rate of 34%.

The duration of the interviews was on average between 50 to 60 min and were all conducted over a period of 4 months, from December 2020 to March 2021, by videoconference. All interviews were audio-recorded, transcribed, and detailed notes were taken during the course of the interviews. The interview transcripts were reviewed and independently coded. Experts were given an interview guide (either in English or in Spanish) prior to the interview.

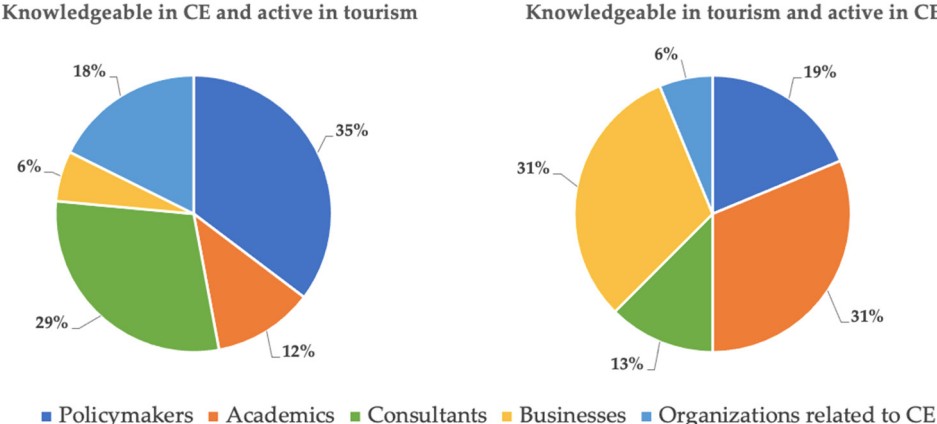

**Figure 1.** Background classification of groups leading the CE transition.

**Table 3.** Interview topics and sample questions.

| Topic | Sample Questions |
|---|---|
| Background information | - First, could you please introduce yourself? Where are you currently working and what is your job position?<br>- How are you related to the topic of circular economy and tourism? |
| Challenges of implementing a CE model in the tourism industry at 3 different levels of abstraction | - In your opinion, what are the challenges that exist at a macroenvironmental level for moving towards circular tourism?<br>- In your opinion, what are the challenges that exist at a microenvironmental level for moving towards circular tourism?<br>- In your opinion, what are the challenges that exist at an organizational level for moving towards circular tourism? |
| CE challenges in tourism under the COVID-19 context | - Do you think that COVID-19 situation has brought new challenges in the implementation of a circular economic model in the tourism sector? If so, what are they? If not, why not? |

This guide provided information about the interviewer, the length of the interview, which means were going to be used to conduct the interview, and the purpose of the former. Furthermore, a brief explanation was given for each level of abstraction (macroenvironmental, microenvironmental and organizational) in order to be aligned on the three different types of questions shown in Table 3 under the topic "challenges of implementing a CE model in the tourism industry at 3 different levels of abstraction". These semi-structured interview questions were derived from the previously conducted systematic literature review [13]. During the course of the interview, the interviewer had all the 68 CECPs available to note down how many were stated and, in case some were overseen, they were asked further questions. To ensure that as much as possible CE challenges were covered, the interviewer had a column next to the questions for each level of abstraction as a reminder, such as this one for the macroenvironmental level: "did he/she mention CE challenges within the following elements? (Political, Economic, Social, Technological, Legal, Environmental)."

Interviewees were given the flexibility to choose the level that they felt more comfortable with according to their expertise (e.g., policymakers mostly preferred talking about the macroenvironmental level). We required the experts to rank the challenges they mentioned in terms of importance and urgency in a 4-point Likert scale -one meaning not important at all/not a priority; two meaning slightly important/a priority; three being moderately important/moderate priority and four meaning very important/high priority-With this ranking we built a decision matrix (see Section 5), considering not only importance and urgency but also considering the variable frequency, which allowed us to find out which are the most crucial challenges.

The criterion followed to determine the key challenges was to choose those with an average score of at least three in importance and urgency and which were mentioned at least by 20% of the experts.

Moreover, we included one last question related to the coronavirus pandemic. Table 3 gives an overview of sample questions per topic discussed during the interviews.

The process of data collection is performed until the point of theoretical saturation is reached [66,67]. We considered that the saturation point was reached at interviewee number 33 as no additional CECP emerged over the course of the last six interviews [68]. Figure 2 shows a graphic representation of the number of CE patterns identified as the number of interviews conducted increases. Once we reached interview 27, a total of 34 CECPs were mentioned. After interviewing six more experts, no new CECP emerged; we decided then to stop the process, considering that the saturation point had been reached with 33 interviews and a total of 34 CECPs mentioned.

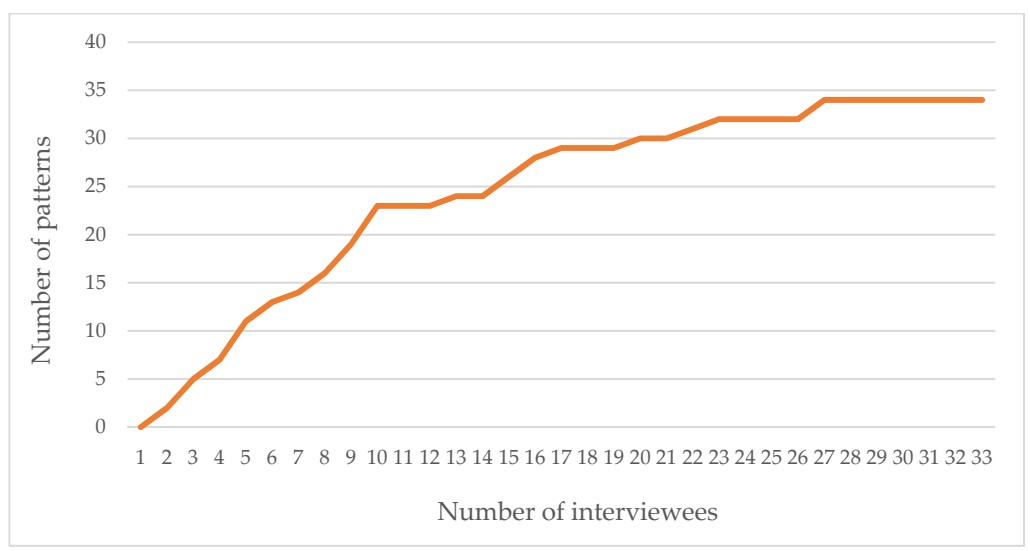

**Figure 2.** Saturation of CECPs mentioned by experts.

*4.2. Data Analysis*

The data analysis was performed in two phases of coding described below. For both phases, in order to ensure reliability between the different coders [69,70], all the coding was performed independently by two researchers following a set of coding rules. The codes were then reviewed, and diverging coding results were solved through mutual discussion sessions. We decided to use manual coding instead of automatic computer coding software, as the entities of codification (challenges mentioned by the interviewees) were clearly delimited, and the level of complexity of the coding was low. Thus, clearly stating the challenges that we had to map to the provisional lists of codes. Additionally, in a third phase, we coded the negative effects of COVID-19 on the identified CECPs.

4.2.1. Phase 1: Identification of CECPs Applicable to the Tourism Industry

The objective for the first phase of coding was to identify which CECPs are applicable to the tourism industry based on the challenges mentioned by the experts interviewed. Given the large number of challenges mentioned (a total of 255), we used a similar approach as Remane et al. [71] to reduce the complexity of coding a large set of data. They coded 487 entities in two steps. First, they applied meta-level coding to cluster the entities. Second, they analyzed each cluster separately for more efficient pattern identification.

Furthermore, Saldaña [72] recommends leveraging provisional lists of codes when previous research in the field already exists. We derived our provisional lists of codes from our previously conducted systematic literature on CECPs (see Section 3). A pattern was identified when at least 2 challenges mentioned by two different interviewees matched with

the code of the same CECP from our systematic literature review [13]. Figure 3 summarizes the approach.

**Figure 3.** Codification approach.

4.2.2. Phase 2: Specificity for Each Identified CECP in the Tourism Industry

Once we have mapped the challenges mentioned by the interviewees to identify which CECPs are applicable to the tourism industry, we had to identify which of the applicable CECPs needed to be further specified for the tourism industry. We argue that a CECP could either need a high or a low specificity. High specificity means that the CECP requires further information to be relevant for the tourism industry. Thus, the CECP, as described in a general way, did not sufficiently capture the emerging ideas of the CE challenges in tourism mentioned by the interviewees. On the other hand, a low specificity means that the CECP, as described in the literature, already captures the emerging ideas from the challenges mentioned by the interviewees. To assess the level of specificity of a CECP for the tourism industry, we leveraged a three-step approach.

(1) First, we leveraged the description of the analyzed CECP from the systematic literature (an example is illustrated in Section 5.1, Table 4, second column "explanation of CECP in the literature"). We used open coding to identify the different ideas that the description encompassed.

(2) Second, we used open coding to converge the challenges mentioned by the interviewees that were mapped to the analyzed CECP to identify the different ideas that the interviews encompassed. We continued converging until we reached the same level of abstraction of the ideas from the CECPs descriptions.

(3) Third, we leveraged the ideas of the CECP description from the systematic literature review as provisional lists of codes to evaluate which ideas from the interviews matched. When more than 50% matched, we defined the CECP as having a low specificity and otherwise as having a high specificity.

**Table 4.** Applicability of CECPs identified in the literature to the tourism industry.

| Abstraction Levels | General CECPs from Literature Review | Tourism CECPs from Expert Interviews | Applicability PERCENTAGE for Tourism |
|---|---|---|---|
| Macroenvironmental | 18 | 12 | 67% |
| Microenvironmental | 14 | 7 | 50% |
| Organizational | 36 | 15 | 42% |
| Total | 68 | 34 | 50% |

The coded ideas where then used to describe the specificity of each CECP for the tourism industry. Examples of CECPs with high and low specificity can be seen in Appendix A (Tables A1–A3).

4.2.3. Phase 3: Identification of Negative COVID-19 Effects for CECPs in Tourism

In addition to phases 1 and 2, we wanted to assess whether the coronavirus pandemic had any effect on previously determined CECPs or whether new challenges needed to be considered.

The interviewees mentioned a total of 59 COVID-19 specific challenges. Similar to phases 1 and 2 combined, we first mapped to which CECP the COVID-19 challenges were related by leveraging the provisional lists of codes described in phase 1. Second, we used open coding to converge the COVID-19 challenges to the same abstraction level as the ideas of the descriptions of the CECPs (same approach as described for the first step of phase 2 in Section 4.2.2). This has allowed us to efficiently summarize the negative effects of COVID-19, clearly mapping their impact to specific CECPs.

**5. Results**

By coding 255 CE challenges mentioned in semi-structured interviews with 33 experts, we were able to make the general research on CECPs relevant for the tourism industry. In this section, we will first present our findings in light of the research questions in general and then provide more details in the following subsections.

Regarding the applicability of the CECPs to the tourism industry (our first research question (i)), we have found out that 34 out of the 68 CECPs apply to the tourism industry. Table 4 below details the number of CECPs per abstraction level, stating the number of general CECPs identified in the previously conducted systematic literature review [13] and the number of tourism-related CECPs identified in this article through the semi-structured interviews.

In total, only 50% of CECPs are applicable to the tourism industry. It can also be observed that the percentage of CECPs applicable to the tourism industry is higher on a macroenvironmental level compared to an organizational level.

Regarding the specificity of the CECPs applicable to the tourism industry (our second research question (ii)), we have found out that 41% of the CECPs needed additional specific information to make the general CECPs relevant for the tourism industry, while 59% of the general CECPs did not require significant additional information to be relevant for the sector. Table 5 states for the three levels of abstraction the number of CECPs with a high and low specificity for the tourism industry.

**Table 5.** Specificity of the applicable CECPs for the tourism industry.

| Abstraction Levels | Tourism CECPs with High Specificity | Tourism CECPs with Low Specificity |
|---|---|---|
| Macroenvironmental | 5 (42%) | 7 (58%) |
| Microenvironmental | 4 (57%) | 3 (43%) |
| Organizational | 5 (33%) | 10 (67%) |
| Total | 14 (41%) | 20 (59%) |

The findings show that, especially on a microenvironmental level, the general CECPs needed to be specified for the tourism industry, while it seems that only one-third of the CECPs on an organizational level needed to be specified.

Furthermore, we have found out that 10 out of the 34 tourism relevant CECPs are crucial to moving toward a circular tourism model (answering our third research question (iii)). Table 6 provides an overview of the number of crucial Tourism CECPs per abstraction level.

The above finding indicates that the experts perceive that the most crucial challenges to tackle to move toward a circular tourism model are located at a macroenvironmental level.

The next three subsections present for each of the three abstraction levels an overview of the corresponding CECPs clearly identifying the most crucial ones.

**Table 6.** Overview of crucial CECPs applicable to the tourism industry.

| Abstraction Levels | Total Number of Tourism CECPs | Number of Crucial Tourism CECPs | Percentage of Crucial Tourism CECPs |
|---|---|---|---|
| Macroenvironmental | 12 | 6 | 50% |
| Microenvironmental | 7 | 3 | 43% |
| Organizational | 15 | 1 | 7% |
| Total | 34 | 10 | 29% |

Furthermore, the fourth subsection explains the identified negative effects of COVID-19 for four CECPs applicable to the tourism industry (answering our fourth research question (iv)).

Beyond this finding section, the three tables (Tables A1–A3) in the Appendix A provide in-depth information on each CECP relevant for the tourism industry per level of abstraction. Table 7 represents an excerpt of the tables in the Appendix A to illustrate how to leverage this additional source of information.

**Table 7.** Excerpt of macroenvironmental level from Table A1.

| Meta Data on CECPs | Explanation of CECP in the Literature | Specificities for the Tourism Industry |
|---|---|---|
| **Title of CECP:** L.03: (Legal 03)—Insufficient implementation of CE regulations<br>**Area:** Legal<br>**Level of specificity:** low<br>**Average importance:** 3.75<br>**Average urgency:** 3.25<br>**Frequency of specific CECP:** 4 experts | There is a lack of regulatory pressures. CE laws are not strong enough; there is no existing tool to analyze the effectiveness of the proposed rules and laws. Most laws are posed with personal opinion rather than technical expertise. There is an inadequate, complex, and fragmented legal system. "Governments and local authorities' responsibilities are not clear on the implementation of CE" [73] (p. 9). Articles: [73–75]. | In order to move tourism businesses toward circularity there needs to be a favorable institutional environment with regulative isomorphic pressures (i.e., laws, sanctions). Tourism companies need a clear indication of what they are allowed to do and what they are not allowed to do with regards to CE. If there are only normative isomorphic pressures, they will not do so, as they are just recommendations that are not legally binding. |

The first column of the tables in the Appendix A captures the metadata for each of the 34 CECPs applicable to the tourism industry. The information included:

- **Title of CECP:** Identifier of the CECP, as well as the corresponding title
- **Area:** Element of the corresponding framework leveraged to group the CECPs
- **Level of specificity:** Whether the specificity is low or high
- **Average importance:** Giving the average of the importance provided by the interviewees
- **Average urgency:** Giving the average of the urgency provided by the interviewees
- **Frequency of specific CECP:** Stating how many experts from the 33 interviewed have named the CECP.

The second column provides the description of the general CECP derived from the literature and references the literature mapped to the CECP [13]. This allows us to understand the CECP that is industry unspecific and refers to the original research for more in-depth information.

The third column complements the explanation of the general CECP by providing additional information that is specific to the tourism industry and has emerged from the expert interviews.

*5.1. Macroenvironmental Level*

The 33 experts interviewed mentioned a total of 90 CE challenges on a macroenvironmental level that were relevant to tourism. Through the codification of these challenges, we were able to group them into 12 CECPs that could be mapped to those previously identified in the literature. Most challenges identified for this level have similar high averages in

terms of importance and urgency but differ in the frequency mentioned by the experts. Figure 4 provides an overview of the macroenvironmental CECPs.

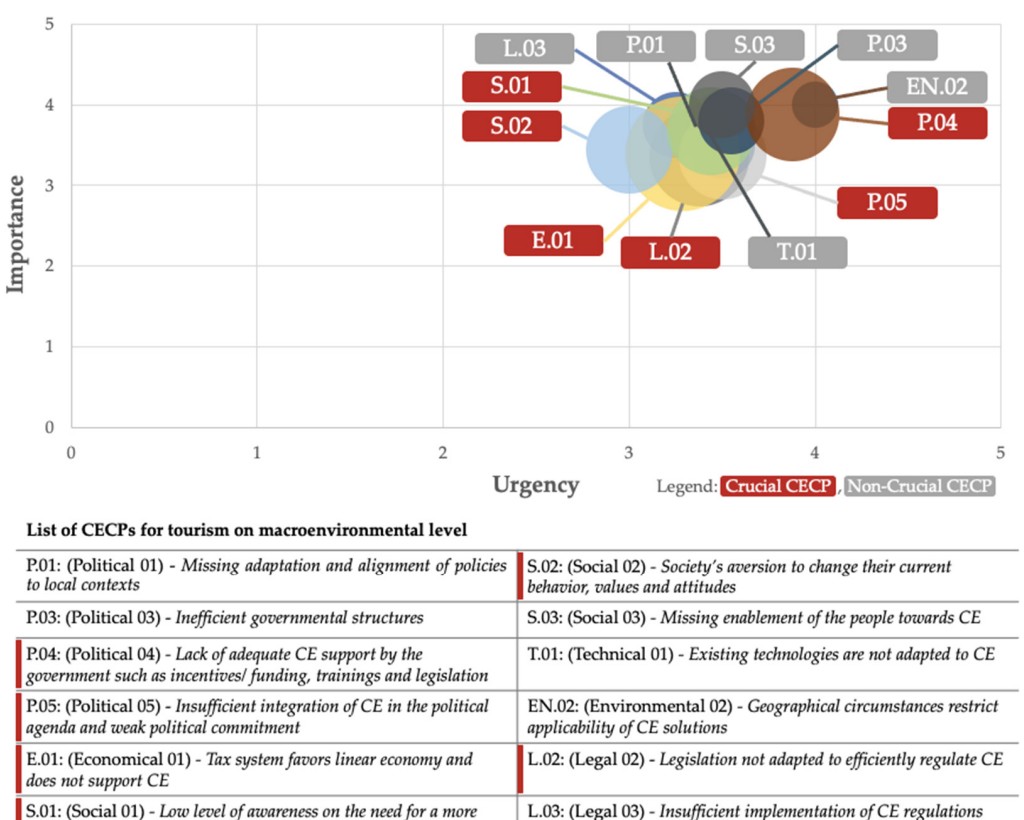

**List of CECPs for tourism on macroenvironmental level**

| | |
|---|---|
| P.01: (Political 01) - *Missing adaptation and alignment of policies to local contexts* | S.02: (Social 02) - *Society's aversion to change their current behavior, values and attitudes* |
| P.03: (Political 03) - *Inefficient governmental structures* | S.03: (Social 03) - *Missing enablement of the people towards CE* |
| P.04: (Political 04) - *Lack of adequate CE support by the government such as incentives/ funding, trainings and legislation* | T.01: (Technical 01) - *Existing technologies are not adapted to CE* |
| P.05: (Political 05) - *Insufficient integration of CE in the political agenda and weak political commitment* | EN.02: (Environmental 02) - *Geographical circumstances restrict applicability of CE solutions* |
| E.01: (Economical 01) - *Tax system favors linear economy and does not support CE* | L.02: (Legal 02) - *Legislation not adapted to efficiently regulate CE* |
| S.01: (Social 01) - *Low level of awareness on the need for a more sustainable economy* | L.03: (Legal 03) - *Insufficient implementation of CE regulations* |

**Figure 4.** Macroenvironmental CECPs.

Figure 4 above is structured as follows. The Y-axis indicates the level of importance, and the X-axis indicates the level of urgency. Each circle represents a CECp. The size of the circle varies depending on the number of times that the CECP was mentioned by the experts. The colors red and gray are used to distinguish the crucial CECPs (red) from the non-crucial CECPs (gray). The legend below provides the full name of the abbreviated CECPs that appear in the graph.

*5.2. Microenvironmental Level*

For this level, a total of 82 CE challenges were mentioned by the experts. Once performed with the codification, we found out that they could be grouped and mapped to seven CECPs in the literature. Figure 5 below provides an overview of the microenvironmental CECPs. Three out of the seven CECPs are considered the most crucial ones for the tourism industry according to the experts. The CECP "R.05: (Resources 05)—Lack of proof of solid CE theory, concepts, methods, measurements and role models (especially business models)" is the one that clearly stands out from the rest, being the most mentioned CECP by the experts (21 out of 33). Furthermore, the other crucial CECPs "VC.03: (Value Chain 03)—Lack of willingness and trust to collaborate across the value chain" and "I.02: (Infrastructure 02)—Inefficient waste management/recycling systems, practices and infrastructures" were mentioned by 18 and 8 experts, respectively. With regards to the latter CECP I.02 (Infrastructure 02), the topic of recycling, one of the last loops from the technical cycle in the EMF circular systems diagram [76], was not mentioned at all in 16 out of the 33 interviews. It was just specifically mentioned by 17 out of the 33 experts, of which 53% insisted that recycling should not be the only option for moving toward CE and that it should not be the first priority for policymakers, businesses, and individuals. According to

them, the challenges around maintaining products in use, ensuring its reuse and remanufacturing have a higher priority. For example, interviewee 9 mentioned "everybody talks about recycling when in fact they don't talk enough about redesigning the materials, so recycling is not important to me compared to redesigning" and interviewee 17 stated "in the end, recyclability is the last thing that you want to do according to circular economy principles, because it's where the most value is lost". The remaining 47% of the experts that mentioned recycling put the emphasis on the lack of infrastructure to do so and the high costs associated with it. For instance, interviewee 21 mentioned "buying recycled bottles is more expensive costing around 3€ than buying virgin glass bottles that cost just approximately 1€" and interviewee 29 stated that "it's too much effort for businesses to recycle even if they want to because the infrastructure is not there".

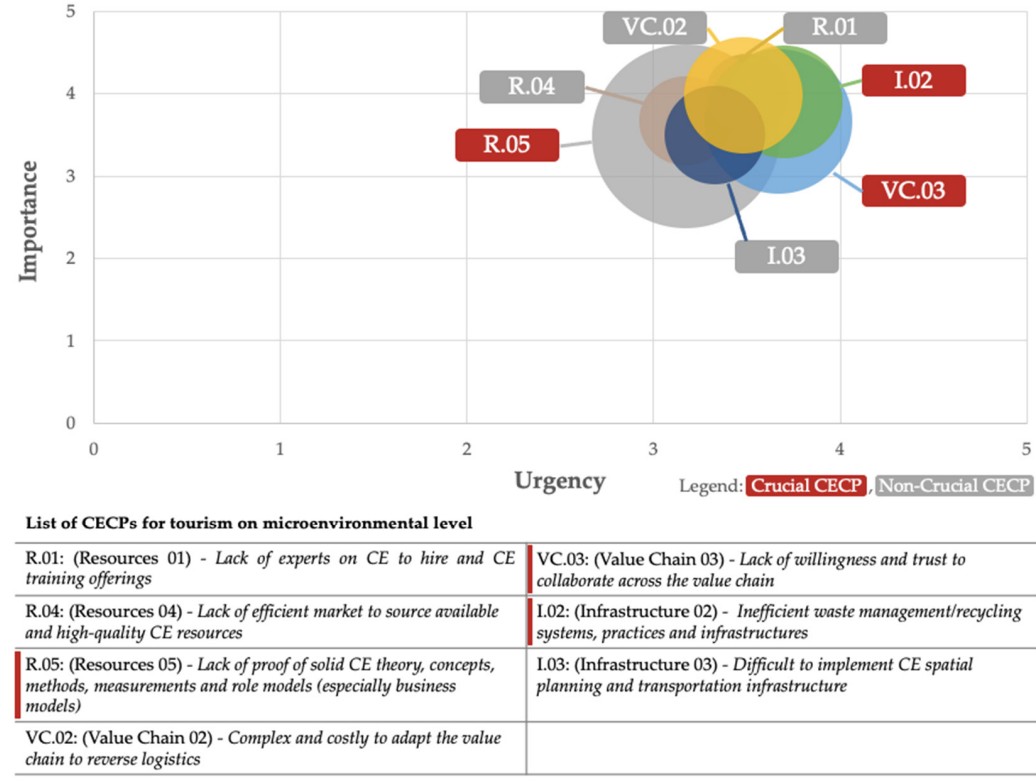

**List of CECPs for tourism on microenvironmental level**

| | |
|---|---|
| R.01: (Resources 01) - *Lack of experts on CE to hire and CE training offerings* | VC.03: (Value Chain 03) - *Lack of willingness and trust to collaborate across the value chain* |
| R.04: (Resources 04) - *Lack of efficient market to source available and high-quality CE resources* | I.02: (Infrastructure 02) - *Inefficient waste management/recycling systems, practices and infrastructures* |
| R.05: (Resources 05) - *Lack of proof of solid CE theory, concepts, methods, measurements and role models (especially business models)* | I.03: (Infrastructure 03) - *Difficult to implement CE spatial planning and transportation infrastructure* |
| VC.02: (Value Chain 02) - *Complex and costly to adapt the value chain to reverse logistics* | |

**Figure 5.** Microenvironmental CECPs.

Please find under the legend of Figure 4 in Section 5.1 the explanation of how the figure is structured.

### 5.3. Organizational Level

Within this level, we identified a total of 83 CE challenges mentioned by the experts that were relevant to tourism. The codification allowed us to group the 83 CE challenges into 15 CECPs that could be mapped to those previously identified in the literature. However, only one crucial CECP was identified. As we can see in Figure 6 below, most of the organizational CECPs are located within the highest levels of importance and urgency; however, we also see some CECPs at the lower end of the importance and urgency spectrum, such as CECP "RS.01: (Revenue Streams 01)—Cannibalization of sales due to new circular products."

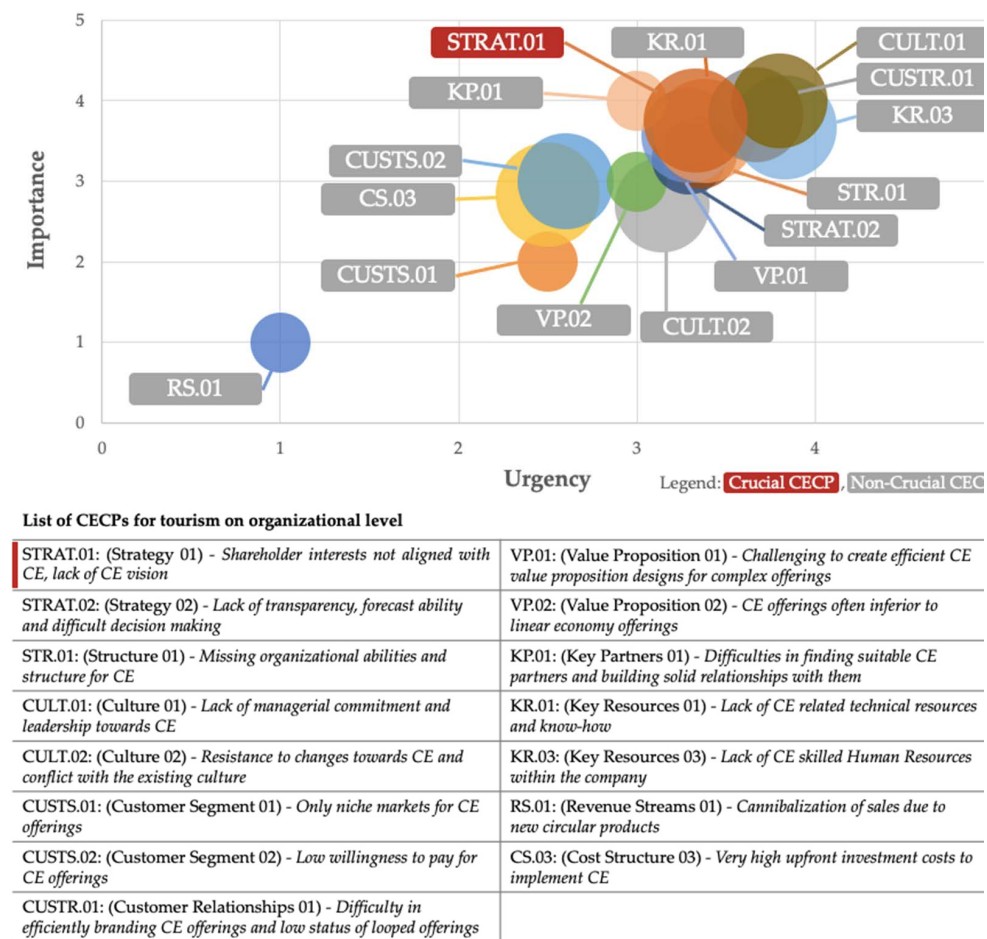

**Figure 6.** Organizational CECPs.

### 5.4. Negative Effects of COVID-19 on Certain CECPs

COVID-19 has not led to the appearance of new CECPs. Instead, we have identified that the negative effects of COVID-19 have affected four CECPs applicable to the tourism industry. In the following, we describe these effects for the four CECPs.

*Special COVID-19 regulations impeding a CE model*—Nine interviewed experts have argued that governments have implemented new regulations to contain the COVID-19 pandemic. These are, for example, the compulsory use of single-use plastic items in the hospitality industry and food and beverage, which has put on hold the progress made toward reusable items. For example, in certain cafés, there is a trend to move away from reusable cups and back to the takeaway model. Even if clients are willing to use reusable cups the cafés will not accept them due to strict COVID-19 protocols in place dictated by the government. This has had a negative effect on the CECP "L.02: (Legal 02)—Legislation not adapted to efficiently regulate CE", as it has made it more difficult to become circular with these new regulations in place.

*Sustainability positions in companies at risk*—From the interviewed experts, three have stated that the tourism industry has faced a serious economic crisis due to COVID-19 and that sustainability positions within organizations were the first ones to be cut. Furthermore, the unemployment rate in the industry has sharply increased. This has resulted in a CE brain drain toward other industries that were less impacted by the pandemic. The described negative effect has impacted the CECP "R.01: (Resources 01)—Lack of experts on CE to hire and CE training offerings" by further diminishing the number of CE experts available for tourism.

*Increase in waste generation*—COVID-19 has given rise to new types of waste. Three of the experts interviewed pointed out that this is also the case in the tourism industry.

In order to ensure a safe and COVID-19 free environment, the industry has been forced to implement a number of measures (such as plastic partitions, gloves, sanitizer gels, disposable masks, etc.), which generate new waste. In addition, experts mention that this waste has not been properly managed, putting further pressure on the environment. This has had a negative effect on the CECP "I.02: (Infrastructure 02)—Inefficient waste management/recycling systems, practices and infrastructures" by not being able to put adequate systems in place that prevent the waste generated from reaching landfill or incineration sites.

*Shift in priorities within organizations*—14 experts have mentioned that environmental issues are not currently a top priority for businesses that are facing serious cash flow problems or on the verge of bankruptcy. They are focusing on survival rather than investing in CE projects that require high levels of capital. The described negative effect has impacted the CECP "STRAT.01: (Strategic 01)—Shareholder interests not aligned with CE, lack of CE vision" as it has made shareholders, that already find it very difficult to move toward a CE model, even more reluctant to do so.

## 6. Discussion

In the following, we will discuss our findings by answering three questions that we outline below.

First, "how can the understanding of the 10 most crucial CECPs help tourism practitioners to move the industry towards CE?" For this, we will introduce the Universal Circular Economy Policy Goals framework of the EMF [77]. This framework proposes five policy goals that act as a roadmap for businesses and governments to align and follow when moving toward a circular economy.

Second, "how should we further develop the research field on CE challenges based on the findings for the tourism industry?" We will answer this question by explaining the applicability and specificity needed to translate the CECP to the tourism industry and by extrapolating those insights to the research field on CECPs in general.

Third, "what can we learn from the negative effects of COVID-19 to make CE endeavors more resilient to future pandemics?" To answer this, we will propose three possible solutions to counteract the probable negative effects of future pandemics.

### 6.1. How can the Understanding of the 10 Most Crucial CECPs Help Tourism Practitioners to Move the Industry toward CE?

Understanding the challenges is especially crucial for policymakers in order to move the tourism industry toward circularity [15]. The EMF proposes five goals that policymakers should work on as a starting point to accelerate the transition to a CE [77]. The goals with their short description and the most crucial CECPs mapped to those goals can be seen in Table 8.

6.1.1. How Pursuing Goal 1, "Stimulate Design for the Circular Economy", Helps Tackle the Crucial CECPs in Tourism

No crucial CECP has been mapped to goal 1. This might be due to the fact that goal 1 has a strong focus on the design of products, while the tourism industry is a rather service-oriented industry. Therefore, we suggest to not focus on this objective when prioritizing.

This being said, some non-crucial CECP in tourism could be linked to this goal. For example, "VP.01: (Value Proposition 01)—Challenging to create efficient CE value proposition designs for complex offerings" could be tackled by more comprehensive product policies that support repairability of textiles in hospitality through the availability of parts to enable repair and provision of repair manuals.

**Table 8.** Mapping of the most crucial CECPs to the five Universal Circular Economy Policy Goals.

| Goals | EMF Descriptions | Mapped Crucial CECPs |
|---|---|---|
| Stimulate design for the circular economy | "How policy can incentivize the switch to circular design practices and circular business models at scale and across sectors" (p. 32). | No crucial CECP has been mapped to this goal |
| Manage resources to preserve value | "Focuses on developing a rich system of resource management that keeps goods and materials in productive use and at high value" (p. 36). | I.02: (Infrastructure 02)—Inefficient waste management/recycling systems, practices and infrastructures |
| Make the economics work | "Focuses on creating the economic conditions needed to scale circular outcomes" (p. 42). | L.02: (Legal 02)—Legislation not adapted to efficiently regulate CE<br>E.01: (Economical 01)—Tax system favors linear economy and does not support CE<br>R.05: (Resources 05)—Lack of proof of solid CE theory, concepts, methods, measurements and role models (especially business models)<br>STRAT.01: (Strategy 01)—Shareholder interests not aligned with CE, lack of CE vision |
| Invest in innovation, infrastructure, and skills | "Focuses on using public finance capabilities to invest in circular economy opportunities and skills and mobilize private investment" (p. 46). | P.04: (Political 04)—Lack of adequate CE support by the government such as incentives/ funding, trainings and legislation |
| Collaborate for system change | "Focuses on the "how" of policymaking for system change—the mechanisms for developing new policies and aligning existing ones in order to unlock a systemic, economy-wide transition to a circular economy" (p. 50). | VC.03: (Value Chain 03)—Lack of willingness and trust to collaborate across the value chain<br>S.01: (Social 01)—Low level of awareness on the need for a more sustainable economy<br>S.02: (Social 02)—Society's aversion to change their current behavior, values and attitudes<br>P.05: (Political 05)—Insufficient integration of CE in the political agenda and weak political commitment |

6.1.2. How Pursuing Goal 2, "Manage Resources to Preserve Value", Helps Tackle the Crucial CECPs in Tourism

Only the crucial CECP in tourism "I.02: (Infrastructure 02)—Inefficient waste management/recycling systems, practices and infrastructures" has been mapped to goal 2. Therefore, it is together with goal 4 the second less critical goal to focus on.

Pursuing goal 2 could help tackle the CECP "I.02 (Infrastructure 02)" for example, by developing and harmonizing collection and sorting policies for hotels to have higher value organic loops; or by leveraging spatial planning policies to enhance material flows and industrial symbiosis, especially when planning new tourism sites and resorts. However, the focus should not be solely on improving recycling infrastructures through downstream innovation to tackle this CECp. It should also be on the redesign of products to avoid waste been generated in the first place (upstream innovation) and on remanufacturing/refurbishing to help keep most of the value of products.

6.1.3. How Pursuing Goal 3, "Make the Economics Work", Helps Tackle the Crucial CECPs in Tourism

Four CECPs in tourism have been mapped to goal 3. This makes it together with goal 5, the ones that tackle most CECPs.

Pursuing goal 3 could help tackle the CECP "L.02: (Legal 02)—Legislation not adapted to efficiently regulate CE" by, for instance, reviewing competition policy to foster collaboration and cooperation for innovation within the tourism value chain.

Furthermore, pursuing goal 3 could help tackle the CECP "E.01: (Economic 01)—Tax system favors linear economy and does not support CE" by aligning taxation with circular economy outcomes, e.g., tax reductions for hotel buildings that are constructed in line with circular economy principles.

Another CECP that could be tackled by pursuing goal 3 is "R.05: (Resources 05)—Lack of proof of solid CE theory, concepts, methods, measurements and role models (especially business models)" by having a detailed taxonomy on CE practices across the tourism sector to ensure greater transparency so that investors have a higher willingness to support CE opportunities in tourism.

Lastly, pursuing goal 3 could help tackle the CECP "STRAT.01: (Strategy 01)—Shareholder interests not aligned with CE, lack of CE vision" by increasing the attractiveness of circular tourism endeavors through, for example, attaching conditions to state aid and government funds (especially funds to support the development of tourism activities in the different regions) to the circularity of business opportunities.

### 6.1.4. How Pursuing Goal 4, "Invest in Innovation, Infrastructure, and Skills", Helps Tackle the Crucial CECPs in Tourism

Only the CECP in tourism "P.04: (Political 04)—Lack of adequate CE support by the government such as incentives/ funding, trainings and legislation" has been mapped to goal 4.

Pursuing goal 4 could help tackle this CECP by helping SMEs (that represent a large fraction of the tourism industry) to obtain easier access to CE funding for example provided by the EU in form of the Horizon2020 fund.

### 6.1.5. How Pursuing Goal 5, "Collaborate for System Change", Helps Tackle the Crucial CECPs in Tourism

For goal 5, four CECPs in tourism were identified, making goal 5 as important as goal 3.

Pursuing goal 5 could help tackle the CECP "VC.03: (Value Chain 03)—Lack of willingness and trust to collaborate across the value chain" by, for example, promoting the establishment of working mechanisms for the highly fragmented number of tourism agents such as creating governmentally supported "transition brokers" (as some interviewed experts called them) such as the European Circular Economy Stakeholder Platform.

Furthermore, the CECPs "S.02: (Social 02)—Society's aversion to change their current behavior, values and attitudes" and "S.01: (Social 01)—Low level of awareness on the need for a more sustainable economy" could be tackled by developing and implementing awareness-raising campaigns as for example placing adds at the airport when tourists wait for their luggage.

Moreover, the CECP "P.05: (Political 05)—Insufficient integration of CE in the political agenda and weak political commitment" could be tackled by accelerating progress through measurement and use of data on the circularity of tourism activities to put pressure on the political parties by informed citizens.

### 6.2. How Should We Further Develop the Research Field on CE Based on the Findings for the Tourism Industry?

The research presented in this article represents the first attempt of its kind to do general research on CECPs relevant for a specific industry. Similar research endeavors for other industries are needed to derive evidence-based implications for the research field on CE. However, the observations within this study can already point us toward potential research implications.

First, it is important to notice that the research on CE challenges, in general, seems quite mature. Even though we have conducted data acquisition until reaching theoretical saturation (see Figure 2 in Section 4.1), no new CECP has been identified.

However, our findings point to the fact that more research is needed to make the general research relevant for different industries. This is illustrated by the finding that for the tourism industry, only 50% of CECPs were applicable (see Table 4 in Section 5), of which 41% had to be further specified to be relevant for the tourism industry (see Table 5 in Section 5). From this, several additional research implications could be derived. Studying the causes for a certain level of applicability and specificity for different industries could help us understand the underlying mechanisms. Without being able to provide evidence, such causes for the tourism industry could be its characteristics differentiating it from

other industries. Such characteristics of the tourism industry could be its seasonality, heterogeneity, intangibility, and absence of ownership. Being aware of such causes could help us to better understand how to make general research beyond challenges relevant for industries. We could derive implications from this, for example, for the adjacent field of research in CE drivers for different industries. We do believe that continuing to strengthen the general research on CE, while making it relevant for the different industries is helpful to build upon a shared understanding of the field and to promote cross-industry learning. We thus believe it would be harmful to strive for industry specific research without the link to the general field.

Furthermore, comparing the findings on the different levels of abstraction (macroenvironmental, microenvironmental and organizational) hints to further implications. It is not surprising that the organizational level, with 42% of CECPs applicable to the tourism industry (see Table 4 in Section 5), has the lowest rate of all three levels. This could be because the challenges on an organizational level are highly dependent on the type of organizations (e.g., business to business versus business to customer companies, service versus product companies) that mainly compose the industry. In contrast, challenges on a macroenvironmental level might be relevant for various industries (e.g., taxes apply to all industries). Thus, we believe it is important, especially for research on an organizational level, to discuss the applicability of findings of CE to other industries. For example, it might be that CE findings for hotels can be applied to other service-oriented organizations while being less applicable for product-oriented organizations.

Moreover, the allocation of the most crucial CECPs for the tourism industry indicates that the macroenvironmental level, with 50% of all crucial CECPs (see Table 6 in Section 5), represents the most important level to address if we want to promote the transition toward a CE. This finding suggests that research should first focus on helping to establish favorable conditions at a macroenvironmental level before focusing on the microenvironmental and organizational levels.

*6.3. What Can We Learn from the Negative Effects of COVID-19 to Make CE Endeavors More Resilient to Future Pandemics?*

As presented in the results (Section 5.4), we have identified negative effects from COVID-19 on four CECPs in tourism mentioned by the experts interviewed. In the following, we propose three solutions that could make the endeavors toward CE more pandemic resilient.

*Establish a CE rescue fund*—From our interviews we have understood that job positions and projects linked to CE tend to be the first ones to be cut when a pandemic translates into an economic crisis. In the case of the tourism industry the pandemic has created an important economic crisis with nearly no tourism activity for months [78]. The industry has reported that many sustainability professionals have left the tourism industry to start new careers in industries less hit by the pandemic. This brain drain has set back many CE endeavors by years. We argue that establishing a CE fund could subsidize sustainability positions within the industry to ensure continuity in the transition toward a more circular economy. This would tackle the negative pandemic effects on CECP "R.01: (Resources 01)—Lack of experts on CE to hire and CE training offerings". A similar logic could be applied to rescue CE projects when companies enter in survival mode by providing funds for CE projects that would be cut. This would tackle the negative pandemic effects on CECP "STRAT.01: (Strategy 01)—Shareholder interests not aligned with CE, lack of CE vision".

*Work toward pandemic regulations that encompass CE principles*—During the COVID-19 pandemic, many regulations have been put in place in an ad-hoc reaction to quickly find solutions to reduce the risks of transmission by increasing hygiene standards. These regulations have rarely encompassed CE principles. For example, in the tourism sector, the accommodation providers and restoration in some countries were forced to wrap certain items with single-use plastics to ensure certain hygiene standards (such as TV remote controls wrapped in plastics and the cutlery vacuum-packed with one-use sanity towels). Developing regulations that can harmonize hygiene and CE considerations is an important

step to counter the negative pandemic effects on CECP "L.02: (Legal 02)—Legislation not adapted to efficiently regulate CE".

*Establish pandemic-reactive waste management systems*—One key aspect that the interviewed experts highlighted is the fact that waste management systems had difficulties changing conditions due to COVID-19. For example, in countries in which the waste management systems are publicly owned, the waste bins in residential areas were overloaded. This can be explained by the fact that more people stayed at home, producing more waste. In contrast, the bins at touristic locations were nearly not in use due to the lack of tourists. Therefore, shifting waste management assets and adapting systems in a more agile way could help us prevent miss-waste management. In countries in which the waste management systems are privately owned, some touristic destinations reported that the waste management systems were simply shut down with a lack of tourism income, as it was not financially viable for the private companies to keep operations running. Thus, establishing a pandemic-reactive waste management system would allow countering the negative pandemic effects on CECP "I.02: (Infrastructure 02)—Inefficient waste management/recycling systems, practices and infrastructures".

## 7. Conclusions

In this paper, we make a new contribution to the study of challenges for the transition toward a CE, the importance of which has been highlighted in several research papers [3,9,10]. Our starting point is our previously published systematic literature review [13] in which we review the highly fragmented literature on CE challenges.

In that paper, we grouped the 731 CE challenges considered by 42 articles into 68 circular economy challenge patterns (CECPs) that were able to capture the core ideas of the CE challenges recurrently mentioned in the literature. To cluster the 68 CECPs in a logic manner, we grouped the CECPs into three levels: (i) a macroenvironmental level leveraging the PESTEL framework; (ii) a microenvironmental level, distinguishing between the areas of resources, value chain, and infrastructure, and (iii) an organizational level, using the "ordering moments", "business model canvas", and "Ellen MacArthur Foundation systems diagram" frameworks. This analysis has not only provided a holistic view of the challenges impeding the transition toward a CE but also allowed, by matching the literature on CE challenges with a set of CECPs, to quickly access all related research on the CECP of interest.

However, further research seems necessary to make general research on CE challenges relevant to different industries [15]. Thus, in this article, we aimed to make the general research on CECPs relevant for the tourism industry [13]. We defined relevance for the tourism industry as understanding which of the CECPs apply to the tourism industry; which need to be further specified beyond the general description of the CECP; which are the most crucial ones for the tourism industry; and which were negatively impacted by the COVID-19 crisis.

By interviewing 33 CE experts (reaching theoretical saturation), we have identified 34 out of the 68 general CECPs that are applicable to the tourism industry. We assessed that 41% of those CECPs had a high specificity for the tourism industry; thus, specific explanations beyond the description of the general CECPs were needed to make those CECPs relevant for the tourism industry. Furthermore, we identified the 10 most crucial CECPs for the tourism industry by leveraging the average importance and urgency of the CECPs, as well as the frequency of the CECP being mentioned by the interviewed experts. Finally, we have identified negative COVID-19 effects for four CECPs.

Our findings are especially interesting for policymakers, as the identification of the 10 most crucial CECPs provides guidance for policymaking to move the tourism industry toward a circular model within the three levels considered (macroenvironmental, microenvironmental and organizational level):

(i)   Policymakers should especially focus on the macroenvironmental level, as six out of the 10 most crucial CECPs have been mapped to this level. Within the political

area, policymakers need to establish adequate CE support to tackle "P.04: (Political 04)—Lack of adequate CE support by the government such as incentives/funding, trainings and legislation" and make CE a more prominent topic within political parties to tackle "P.05: (Political 05)—Insufficient integration of CE in the political agenda and weak political commitment". Within the economic area, policymakers need to rethink the mechanisms of our current tax system to tackle "E.01: (Economic 01)—Tax system favors linear economy and does not support CE". Within the social area, policymakers need to invest in CE marketing campaigns to tackle "S.01: (Social 01)—Low level of awareness on the need for a more sustainable economy" and "S.02: (Social 02)—Society's aversion to change their current behavior, values and attitude". Within the legal area, legislation need to be reviewed to tackle "L.02: (Legal 02)—Legislation not adapted to efficiently regulate CE".

(ii) The second most important level to focus on is the microenvironmental level that encompasses three out of the 10 most crucial CECPs for the tourism industry. Especially, knowledge on how to efficiently transition toward and operate in a CE needs to be further developed and made available within the industry to tackle "R.05: (Resources 05)—Lack of proof of solid CE theory, concepts, methods, measurements and role models (especially business models)". Furthermore, policymakers need to promote collaboration between the players in the tourism value chain to tackle "VC.03: (Value Chain 03)—Lack of willingness and trust to collaborate across the value chain". Additionally, a solid waste management infrastructure is needed for the tourism industry to tackle "I.02: (Infrastructure 02)—Inefficient waste management/recycling systems, practices and infrastructures".

(iii) At this stage of the transition toward a circular tourism model, the organizational level is perceived as the least important to focus on by the interviewees as only one of the 10 crucial CECPs for the tourism industry has been mapped to this level. However, we believe that this CECP is a crucial one that should not be overlooked. Policymakers need to evaluate through which mechanisms it can be attractive for tourism mechanisms to move toward a circular tourism model to tackle "STRAT.01: (Strategic 01)—Shareholder interests not aligned with CE, lack of CE vision".

Our discussion chapter has put our findings into context. First, we outlined which of the five Universal Circular Economy Policy Goals proposed by the Ellen MacArthur Foundation [77] would help tackle which crucial CECPs in what way for the tourism industry. Second, we discussed how the research field on CE should be further developed, stressing the importance to assess especially the applicability and specificity of CECPs for different industries to make the research more relevant. Third, we have proposed three solutions (derived from the COVID-19 effects on the CECPs in tourism) to make the endeavors toward CE more pandemic resilient for the future.

The contribution of our article is two-fold. From an academic perspective, we have significantly expanded the research on CE challenges in tourism. While previous studies [20,21] have identified challenges mapped to 6 CECPs in total, our research has expanded this field by 28 CECPs for the tourism industry. Furthermore, the endeavor to make the general research on CECPs relevant for the tourism industry has a pioneering character. To the best of our knowledge so far, no such research has been undertaken in the nascent research field of CE challenges. We hope our article can serve as a reference for similar endeavors to make the CECPs relevant for other industries.

On the other hand, the extensive presentation of the CECPs for the tourism industry (included in the tables of the Appendix A) supports practitioners to accelerate the transition toward a circular tourism model. Not only do we state the most crucial challenges to tackle in this journey, but also detail for each of the 34 CECPs of the tourism industry which researchers have worked previously on studies related to the CECp. We thus believe it to be a powerful tool, not only providing an overview of the full spectrum of CECPs in the tourism industry, but also providing depth for each CECP as practitioners can easily related to further studies for each CECp. Furthermore, the analysis of COVID-19 effects

on CECPs in the tourism industry allows practitioners to develop measures to make the efforts toward a more circular tourism model resilient to future pandemics.

This article is not free of limitations. First, we would like to highlight possible limitations regarding the data collection process. Semi-structured interviews, as leveraged in this research, might unwillingly condition the interviewees to respond in a certain manner following the semi-structured interview guideline. Future research might leverage more open questions with various iterations following, for example, a Delphi study to assess whether additional CECPs can be identified. Furthermore, as mainly the network of the authors was leveraged to acquire the interviewees, the selection of experts might have a regional bias. Well aware of this bias risk, we have intended to reach out to experts outside of the regional network of the authors. Indeed, 54% of interviewed experts originate from outside of Spain. However, with only 21% of interviewees coming from outside of the European Union, we argue that further research might be needed to expand our sample size further including experts from other regions.

Second, the research design of codification might represent a limitation. We have decided to leverage a pre-defined list of codes as proposed by Saldaña [72] to ensure the link to previously conducted research on CECPs. However, an open-coding approach might have led to the discovery of new CECPs. Even though the codes stand on a solid foundation as they emerged from an extensive systematic literature review [13], new sources of challenges could lead to the emergence of new CECPs. For example, technological developments as artificial intelligence leveraging the increasing amount of data produced in the tourism industry might create new challenges that have not been studied before. Future research could thus try to identify new CECPs by specifically studying around sources of potentially new CECPs and leveraging rather an open-coding approach.

Third, the design of the data analysis has allowed to efficiently study the large amount of 33 interviewees. More in-depth statistical analysis might reveal additional insights, such as structurally different perspectives among groups of interviewees. Future research could further explore the sample of conducted interviews to better understand the challenges posed by the transition to the CE.

**Author Contributions:** Conceptualization, J.M.-C. and F.L.-d.-P.; methodology, J.M.-C. and F.L.-d.-P.; validation, F.L.-d.-P.; formal analysis, J.M.-C.; investigation, J.M.-C.; data curation, J.M.-C.; writing—original draft preparation, J.M.-C. and F.L.-d.-P.; writing—review and editing, J.M.-C. and F.L.-d.-P.; visualization, J.M.-C. and F.L.-d.-P.; supervision, F.L.-d.-P.; project administration, J.M.-C. and F.L.-d.-P.; funding acquisition, F.L.-d.-p. All authors have read and agreed to the published version of the manuscript.

**Funding:** The article processing charges of this article were funded by the Department of Applied Economics Analysis and the University Institute of Tourism and Sustainable Economic Development, University of Las Palmas de Gran Canaria (Spain).

**Institutional Review Board Statement:** Not applicable.

**Informed Consent Statement:** Not applicable.

**Data Availability Statement:** "The data is not publicly available as the interviewees want to remain anonymous".

**Conflicts of Interest:** The authors declare no conflict of interest. The funder (Department of Applied Economics, University of Las Palmas de Gran Canaria) had no role in the design of the study; in the collection, analyses, or interpretation of data; in the writing of the manuscript; or in the decision to publish the results.

## Appendix A

We include in this Appendix A the three tables previously mentioned in the results (Section 5). Tables A1–A3 provide in-depth information on each CECP relevant for the tourism industry per level of abstraction.

**Table A1.** CECPs on a macroenvironmental level.

| Meta Data on CECPs | Explanation of CECP in the Literature | Specificities for the Tourism Industry |
|---|---|---|
| **Title of CECP:** P.01: (Political 01)—Missing adaptation and alignment of policies to local contexts<br>**Area:** Political<br>**Level of specificity:** High<br>**Average importance:** 3.75<br>**Average urgency:** 3.50<br>**Frequency of specific CECP:** 4 experts | There is a lack of harmonization between the policies applied at an international, national, and local level. The regional policies are not adapted to the local contexts and the sustainability goals of cities are sometimes in conflict with national priorities. "Regional policies not calibrated to local contexts" [79] (p. 7). Articles: [79–83]. | Tourism is characterized by highly heterogeneous offerings highly dependent on the regional and municipal characteristics and on their degree of vulnerability (balance between the economic activity and the environmental pressure). CE policies need to be adapted to the regions and municipalities. To illustrate one example mentioned, if you have a region that is characterized by nomadic tourism and the waste collection infrastructure is underdeveloped, then the CE policies should include a ban on certain disposable items, such as plastics bottles, which are typically associated with nomadic activities. |
| **Title of CECP:** P.03: (Political 03)—Inefficient governmental structures<br>**Area:** Political<br>**Level of specificity:** Low<br>**Average importance:** 3.75<br>**Average urgency:** 3.50<br>**Frequency of specific CECP:** 4 experts | The bureaucracy blocks company's application of sustainability policies and legislations. There is a lack of decentralization of decision-making and lack authority to effect change. "Cities lack the institutional capacity to deliver looping actions across resource types" [80] (p. 11). Articles: [79,80,82,84–90]. | The focus was primarily on the unnecessary bureaucratic procedures that put the tourism industry to a halt when moving toward circularity. No tourism specifications were highlighted by the experts. |
| **Title of CECP:** P.04: (Political 04)—Lack of adequate CE support by the government such as incentives/ funding, trainings and legislation<br>**Area:** Political<br>**Level of specificity:** High<br>**Average importance:** 3.88<br>**Average urgency:** 3.88<br>**Frequency of specific CECP:** 8 experts | Lack of CE support by the government in the form of few financial incentives, training, CE policies (such as for public procurement). "The lack of funding opportunities likely relates to the unclear market demand for CBMs" [91] (p. 6). Articles: [11,12,15,73,74,79,80,84,86,87,91–96]. | Experts insisted that the most important challenge linked to government support is the lack of financial incentives established for tourism organizations (in particular SMEs) to become more circular. One specific point mentioned is that, specifically for the case of the EU, there are a lot of funds available for becoming more circular (such as Horizon2020, COSME), but the SMEs do not have the time and the expertise to understand the complex process for applying to these funds. So even though funds are theoretically available, they are practically not reachable for tourism SMEs, which represent the majority of tourism businesses. Additionally, one expert argued that the government should force OTAs (Online Travel Agencies) by legislation to show clearly the level of circularity of different offers (e.g., Booking.com could be obliged to show the water consumption levels of hotels per year, how hotels and apartments manage their waste, etc.) This would put pressure on the accommodation businesses to become more circular while ensuring transparency for the tourists caring about circular offers. |

**Table A1.** *Cont.*

| Meta Data on CECPs | Explanation of CECP in the Literature | Specificities for the Tourism Industry |
|---|---|---|
| **Title of CECP:** P.05: (Political 05)—Insufficient integration of CE in the political agenda and weak political commitment<br>**Area:** Political<br>**Level of specificity:** Low<br>**Average importance:** 3.37<br>**Average urgency:** 3.50<br>**Frequency of specific CECP:** 7 experts | The silo-mentality within governments hinders the implementation of a circular economy. Lack of strong policy maker's commitment and support for sustainability issues. There is a lack of integrated approach for policymaking and deficient institutional frameworks. Energy-saving and pollution reduction conflicts with GDP due to limited attention by national and regional governments. "Lack of political initiatives supporting CE tourism innovation" [20] (p. 3). Articles: [6,11,20,73,74,79,80,84,88,96–99]. | The only specificity mentioned is that politicians are not fully aware of the impacts caused by the tourism industry, which in turn does not create the willingness to focus on the tourism industry to become more circular. |
| **Title of CECP:** E.01: (Economical 01)—Tax system favors linear economy and does not support CE<br>**Area:** Economic<br>**Level of specificity:** High<br>**Average importance:** 3.37<br>**Average urgency:** 3.31<br>**Frequency of specific CECP:** 12 experts | Tax systems are not aligned with CE (e.g., high taxes on waste, lack of taxation of labor rather than raw materials . . . ). This unfavorable tax environment leads companies to avoid the implementation of CE, although they are willing to do so. "Existing taxation systems, policies as well incentives, are not aligned with the adoption of the CE paradigm" [100] (p. 7403). Articles: [20,73,79,86,91,97,98,100]. | Experts highly criticized the current linear tax systems that are affecting the possibilities to circulate the tourism industry. First, regarding transportation, experts pointed out that taxation on aviation fuels is flawed, because the externalities are not internalized. The problem is that if taxes would be adequately applied to energy, tourism would decrease, and communities highly dependent on tourism would be worse-off and oppose. Second, other experts mentioned that it is cheaper to incentivize linear behaviors than circular ones (i.e., throwing the organic waste to landfill instead of sorting it to incorporate it back to the tourism value chain in the form of compost). Another example mentioned was related to secondhand furniture being more expensive than brand new. This makes it prohibitive to switch toward circular economy procurement strategies, thus demotivating tourism stakeholders to be circular, and puts a lot of environmental pressure by incentivizing a "take-make-waste" model. Third, even though implementing a green tourism tax (direct or indirect) to the tourists can be a great way to compensate the negative impacts created and promote a different kind of tourism with a higher purchasing power, it is perceived as detrimental for the tourism sector in some destinations such as the Canary Islands, as it can affect the path dependency of mass tourist customers, which are an important segment of the tourists coming. Fourth, taxes need to stop being visitor volume growth-oriented to incentivize a certain type of tourism that is better for the environment and for the economy (longer stays), e.g., cycle tourism. |

**Table A1.** *Cont.*

| Meta Data on CECPs | Explanation of CECP in the Literature | Specificities for the Tourism Industry |
|---|---|---|
| **Title of CECP:** S.01: (Social 01)—Low level of awareness on the need for a more sustainable economy<br>**Area:** Social<br>**Level of specificity:** High<br>**Average importance:** 3.67<br>**Average urgency:** 3.44<br>**Frequency of specific CECP:** 7 experts | Consumers are not aware of the importance of CE, which makes it difficult to adopt sustainable practices. Consumers do not see the urgency of changing their habits for the benefit of the environment, society, and the economy. In addition, the awareness among logistic companies and producers is still too limited to trigger a large-scale shift toward a CE. "Need to raise awareness on the impact of 'habitual choices' on environmental, social, political and cultural system" [81] (p. 2201). Articles: [11,15,73,74,79–84,86,95,96,101–103]. | Lack of awareness on CE among tourists and employees in the tourism sector is seen as a challenge when moving toward CE. On the one hand, tourists are less aware of their behaviors when on holidays as to when they are at home. Therefore, it is crucial to raise the awareness of tourists to change their customer travel and consumption patterns toward more circular ones, particularly in developed places with a good waste infrastructure, to fight the core problem of overconsumption. Because it does not impact us in the same way, as we cannot see the big picture of the amounts of waste generated by our consumption and travel habits. On the other hand, employees in the tourism sector have low levels of awareness and understanding, varying between developed and developing countries, on the opportunities and benefits that circularity can bring in tourism and how it can be implemented. For example, in some countries such as South-East Asia, using single-use plastics is preferable as it is a sign of wealth no matter the environmental impacts it can cause related to, e.g., its improper disposal due to deficient waste management infrastructures. |
| **Title of CECP:** S.02: (Social 02)—Society's aversion to change their current behavior, values and attitudes<br>**Area:** Social<br>**Level of specificity:** High<br>**Average importance:** 3.44<br>**Average urgency:** 3.00<br>**Frequency of specific CECP:** 7 experts | There is a rigidity in consumer behavior toward change in their habits. The existing values, norms, and lifestyles may hinder the implementation of a CE, as there is little or no willingness to change their behavior and consumption patterns; customers usually question the quality, health, and safety of reused and remanufactured products and tend to have the wrong perceptions on them. Hence, this lack of willingness to buy used products forces the remanufacturers to not go for refurbishing/remanufacturing. "Lack of customer interest in the environment" [94] (p. 1055). Articles: [6,20,74,79,80,83,88,93,94,98,99,101,102,104,105]. | In general, tourists tend to care less and tend to leave behind their "circular" attitudes when on holidays. Experts have mentioned two major issues that explain this. The first one is the "convenience factor" when traveling, particularly with hand luggage, which is a priority over everything else. The second one is the difficulties in adapting to the different waste systems at every destination they go due to the lack of information, which leads, in turn, to less interest in its proper disposal. For example, in one city, plastics go in one place and paper goes in another place, but it might be that because of the production system at the holiday destination that they put plastic and paper together. |

**Table A1.** *Cont.*

| Meta Data on CECPs | Explanation of CECP in the Literature | Specificities for the Tourism Industry |
|---|---|---|
| **Title of CECP:** S.03: (Social 03)—Missing enablement of the people towards CE<br>**Area:** Social<br>**Level of specificity:** Low<br>**Average importance:** 4.00<br>**Average urgency:** 3.50<br>**Frequency of specific CECP:** 4 experts | There is a lack of understanding of CE among many players in society due to education deficiency on CE. Waste topic is not included sufficiently in school curricula, hindering the enablement of children from taking more circular actions. Low rates of recycling in society are related to a lack of proper education on environmental issues. "Lack of availability of environmental management programs and facilities both under governmental bodies and at academic institutions" [73] (p. 10). Articles: [73,75,79,90,98,99,106]. | The experts emphasized the need to implement system thinking approaches through education of the different stakeholders involved in tourism. No tourism specifications were highlighted by the experts. |
| **Title of CECP:** T.01: (Technical 01)—Existing technologies are not adapted to CE<br>**Area:** Technological<br>**Level of specificity:** Low<br>**Average importance:** 3.50<br>**Average urgency:** 3.50<br>**Frequency of specific CECP:** 4 experts | Technology for CE is not available at scale at a cost-effective level. There is currently limited proof for CE technology. There are many technological limitations for the tracking of recycled materials due to the increasing complexity of products, which make them effective and efficient recovery and reuse of products and components a massive challenge. "Lack of adequate technologies used in landfilling and incineration activities cause huge irrevocable environment losses" [73] (p. 11). Articles: [6,73–75,79,83,84,86,93,97,98,101]. | This challenge was tackled by focusing on the lack of technologies (such as blockchain and big data) to implement selective waste sorting in origin in order to leverage the organic waste generated across the tourism value chain for other uses. |
| **Title of CECP:** EN.02: (Environmental 02)—Geographical circumstances restrict applicability of CE solutions<br>**Area:** Environmental<br>**Level of specificity:** Low<br>**Average importance:** 4.00<br>**Average urgency:** 4.00<br>**Frequency of specific CECP:** 2 experts | Due to the geographical circumstances, the ability to implement certain circular economy procedures may be hampered. "The difference between geographical circumstances affects metabolic flows and applicability of solutions" [83] (p. 7). Articles: [83,101]. | Experts have mentioned that, indeed the geographical circumstances play an important role when implementing CE solutions. For example, in the case of islands, it is more complex due to their limited dimensions that impede economies of scale compared to other continental regions. No tourism specifications were highlighted by the experts. |

**Table A1.** *Cont.*

| Meta Data on CECPs | Explanation of CECP in the Literature | Specificities for the Tourism Industry |
|---|---|---|
| **Title of CECP:** L.02: (Legal 02)—Legislation not adapted to efficiently regulate CE<br>**Area:** Legal<br>**Level of specificity:** Low<br>**Average importance:** 3.36<br>**Average urgency:** 3.36<br>**Frequency of specific CECP:** 9 experts | Existing obstructing and inconsistent laws and regulations hamper circular practices. Service providers cannot legally retain ownership of a sold product, which makes it difficult to implement CE. Existing laws in waste management in some systems do not fit CE concepts. There is a lack of supporting government legislation with inadequately defined multi-level regulatory frameworks favoring linear processes. Legislation hinders circular business models, e.g., legislation on sales of waste materials and on cross-border movement of products for reuse. "Competition legislation inhibits collaboration between companies" [84] (p. 7). Articles: [6,11,15,74,79,80,82,84–86,91,93,97,98,107]. | According to the experts, existing regulations are interfering in the circular transformation of the tourism sector as the legal models are designed to favor the continuation of a linear economic model. For example, when it comes to the sharing of services/assets, i.e., hotel halls, hotel managers face legal barriers to do so. Another example is that certain regulations impede the reuse of materials as well as the end-of-life treatment of the waste to reincorporate appropriately into the supply chain. Furthermore, experts consider that there should be legal incentives to push citizens to choose the best environmental means of transportation, not only reflected in the price but through positive reward measures in place that induce more circular behaviors. For example, an application for the smartphone that can check in real-time the $CO_2$ emissions of the user when deciding to use different means of transport in the region. If you stay below your monthly rate of $CO_2$ emissions, you can convert the non-$CO_2$ emissions in local currencies to purchase local food, etc.<br><br>**Negative COVID-19 effect:** COVID-19 has implemented new regulations that strengthen the current linear models due to health and safety reasons. These are for example the compulsory use of single-use plastic items in the hospitality industry, F&B, which has put on hold the progress made toward reusable items. For example, in certain cafés, there is a trend to move away from reusable cups and back to the takeaway model. Even if clients are willing to use reusable cups the cafés will not accept them due to strict COVID-19 protocols in place dictated by the government. **(Frequency: 9 experts mentioned this)**. |

**Table A1.** *Cont.*

| Meta Data on CECPs | Explanation of CECP in the Literature | Specificities for the Tourism Industry |
|---|---|---|
| **Title of CECP:** L.03: (Legal 03)—Insufficient implementation of CE regulations<br>**Area:** Legal<br>**Level of specificity:** Low<br>**Average importance:** 3.75<br>**Average urgency:** 3.25<br>**Frequency of specific CECP:** 4 experts | There is a lack of regulatory pressures. CE laws are not strong enough; there is no existing tool to analyze the effectiveness of the proposed rules and laws. Most laws are posed with personal opinion rather than technical expertise. There is an inadequate, complex, and fragmented legal system. "Governments and local authorities' responsibilities are not clear on the implementation of CE" [73] (p. 9). Articles: [73–75]. | In order to move tourism businesses toward circularity, there needs to be a favorable institutional environment with regulative isomorphic pressures (i.e., laws, sanctions). Tourism companies need a clear indication of what they are allowed to do and what they are not allowed to do with regard to CE. If there are only normative isomorphic pressures, they will not do so, as they are just recommendations that are not legally binding. |

**Table A2.** CECPs on a microenvironmental level.

| Meta Data on CECPs | Explanation of CECP in the Literature | Specificities for the Tourism Industry |
|---|---|---|
| **Title of CECP:** R.01: (Resources 01)—Lack of experts on CE to hire and CE training offerings<br>**Area:** Resources<br>**Level of specificity:** Low<br>**Average importance:** 4.00<br>**Average urgency:** 3.50<br>**Frequency of specific CECP:** 4 experts | There is not enough qualified workforce on CE. There is a lack of interest and understanding to apply CE across value chains. There is a need for training and education on CE. There is no official training available for employees in repair/refurbish and no guidelines for third-party repair companies. "Lack of qualified personnel in environmental management" [12] (p. 164). Articles: [12,73,91,94,95]. | The tourism industry lacks education in the circular economy as well. No tourism-specific aspect has been mentioned by the experts.<br><br>**Negative COVID-19 effect:** The tourism industry has faced a serious economic crisis due to COVID-19. Experts argue that sustainability positions within organizations were the first ones to be cut. Furthermore, the unemployment rate in the industry has sharply increased. This has resulted in a CE brain drain toward other industries that were less impacted by COVID-19. **(Frequency: 3 experts mentioned this)**. |
| **Title of CECP:** R.04: (Resources 04)—Lack of efficient market to source available and high-quality CE resources<br>**Area:** Resources<br>**Level of specificity:** High<br>**Average importance:** 3.67<br>**Average urgency:** 3.17<br>**Frequency of specific CECP:** 5 experts | There is limited availability and quality of recycling materials due to technological limitations for recycling, product design, and other processes. It is difficult to supply recycled/reused/refurbished products as there is limited demand for looped products. Lack of standardization on refurbishment products leads to a reduced quality. "Lack of market for recycled materials (e.g., glass, polymers)" [86] (p. 34) and "original spare parts are difficult or impossible to attain or have to be transported over long distances" [91] (p. 10). Articles: [15,74,79,80,84,86,91,95,106–108]. | The insufficiently developed supplier market for circular economic products as well plays an important role in the tourism industry. Especially, as the industry is characterized by a high scale demand of branded supplies. E.g., Hotels stated that they cannot purchase bamboo toothbrushes, as the provider are too small to provide the needed number for a hotel chain as well as they are not able to brand such a toothbrush, which is essential for a hotel chain. |

**Table A2.** *Cont.*

| Meta Data on CECPs | Explanation of CECP in the Literature | Specificities for the Tourism Industry |
|---|---|---|
| **Title of CECP:** R.05: (Resources 05)—Lack of proof of solid CE theory, concepts, methods, measurements and role models (especially business models)<br>**Area:** Resources<br>**Level of specificity:** High<br>**Average importance:** 3.48<br>**Average urgency:** 3.17<br>**Frequency of specific CECP:** 21 experts | There is a lack of data and indicators to measure (long-term) benefits of CE activities. Lack of clear, reliable standards to assess CE processes, activities, and materials, leading to lack of public awareness and lack of demand for sustainable products. There is an absence of perceived need to move toward CE "many companies do not see how lifecycle thinking can be applied to their specific operations – or even the benefits of doing so. Many potential users are unaware of how life-cycle approaches can aid in decision making" (p. 20). There is limited awareness of successful CE business models in resource management and planning projects, as well as lack of successful business models to implement CE in supply chain. "Knowledge development in the field of circular business models is still in its infancy" [103] (p. 14). Articles: [11,15,74,75,79,80,84,86,89,91,92,96,99–101,103,106,107,109]. | For the tourism industry, the experts put a strong emphasis on collecting data about the circularity of the industry. Especially as the industry has the potential to be truly digital-enabled, allowing better communication. Some experts suggest, for example, that hotels should track the water consumption of tourists and communicate this to them. However, too many different measures exist within the tourism sector and standardization of circular economy measurement is needed. Experts emphasized that the industry needs to become more transparent about material flows with the help of data (how many materials come in, what materials come in, what materials remain, what materials become stock, what materials become buildings, what waste is generated), to circularize it. Another often cited challenge was having a specific international certification on CE for the tourism sector that is not seen as another "green certification" but as one with a solid international reputation that can be recognized and valued by all the stakeholders in tourism. |
| **Title of CECP:** VC.02: (Value Chain 02)—Complex and costly to adapt the value chain to reverse logistics<br>**Area:** Value Chain<br>**Level of specificity:** High<br>**Average importance:** 4.00<br>**Average urgency:** 3.50<br>**Frequency of specific CECP:** 4 experts | The exchange of materials is limited by the capacity of reverse logistics. The reverse logistics organization and stability prevent companies from implementing circular business models. "The quality, access and attractiveness of recovered products and materials" [97] (p. 7) remains challenging. "Extending the supply chain to include remanufacturing, recycling, repair and refurbishing creates an additional level of complexity, leading to potentially negative impacts in quality, cost, and delivery times" [86] (p. 22). Articles: [11,84,86,97,104,106–108]. | Tourism practices make reverse logistics very difficult. It is cheaper for tourists to purchase many items at the destination than to bring them from their country of origin, to use them during their stay, and then throw them away. This makes it difficult for the tourism industry to establish reverse logistic models. If there would be a proper reverse logistics infrastructure in place, the goods would be collected, transported to a central location, and sorted for reuse, refurbish, or recycling purposes. So, the challenge is two-fold: it requires the involvement of the tourists to return end-of-use products and it requires businesses to look at their wider business model to make sure that products and materials can be reused, remanufactured, repaired, or recycled. |

| Meta Data on CECPs | Explanation of CECP in the Literature | Specificities for the Tourism Industry |
|---|---|---|
| **Title of CECP:** VC.03: (Value Chain 03)—Lack of willingness and trust to collaborate across the value chain<br>**Area:** Value Chain<br>**Level of specificity:** High<br>**Average importance:** 3.67<br>**Average urgency:** 3.67<br>**Frequency of specific CECP:** 13 experts | Network collaboration across the value chain (VC) to facilitate the implementation of CE all along remains very complex. It is very difficult to find and create the appropriate, trustworthy networks (especially from the supply chain) necessary for circularity. "From a supply perspective, a major challenge seems to be the absence of 'green' suppliers for specific inputs that the SME needs in the production process of a product or a service. According to the SMEs, in most cases, markets for these inputs are absent or insufficiently developed in the supply chain. Also, some SMEs report difficulties in implementing a green solution since they are locked in at the bottom of the supply chain or they are part of global supply chains sectors with correlated high environmental impact" [87] (p. 10).<br>It is complicated to have a strong commitment toward the implementation of CE and to get the entire industry on board, as not all, e.g., packaging component manufacturers, packaging equipment users, material producers, waste recovery facilities, have the same interests. "Involves actors from across society and creation of suitable collaboration and exchange patterns" [106](p. 1033) and "CBM is based on collaboration, and that requires trust between parties" [107] (p. 5). Articles: [11,15,74,80,86,87,91,93,101,103,106,107,110,111]. | For the tourism industry, this is clearly the case, as there is a great number of agents across the tourism value chain, which is very extensive and fragmented. Many experts have insisted on the complexity of applying systems thinking in the tourism industry due to the many stakeholders involved. Furthermore, the highly fragmented and extensive tourism industry makes it even more difficult to achieve a coordinated vision to circulate the sector as there is barely any cross-sectoral collaboration. For instance, it is possible to find solutions for the circularity of food, circularity of plastics, but not necessarily within the entire sector. To tackle this missing collaboration and coordination across the tourism value chain, experts have proposed that "transition brokers" are needed to make the transition effect when moving toward CE. |

**Table A2.** *Cont.*

| Meta Data on CECPs | Explanation of CECP in the Literature | Specificities for the Tourism Industry |
|---|---|---|
| **Title of CECP:** I.02: (Infrastructure 02)—Inefficient waste management/recycling systems, practices and infrastructures<br>**Area:** Infrastructure<br>**Level of specificity:** Low<br>**Average importance:** 3.90<br>**Average urgency:** 3.70<br>**Frequency of specific CECP:** 8 experts | Lack of economies of scale in waste treatment/recycling hinders the implementation of appropriate infrastructure necessary for CE, as "it is prohibitively costly for individual organizations to invest in smart enabling technologies for waste management" [75] (p. 19). Furthermore, some regions cannot reach economies of scale, as there is not enough amount of waste and also due to the geographic conditions, such as islands with certain conditions that limit their possibilities as isolated environments, "the bulk density of the roasted material makes it difficult to transport and store it from an economic point of view" [90] (p. 4). Not all regions have the necessary waste containers in public spaces for appropriate waste separation and "points for separated waste collection frequently becoming wasted areas (illegally dumped litter near the separate collection bins)" [79] (p. 8). Non-integrated poor waste infrastructure and long distances between waste generation and treatment. "Dual waste system (households/industrial) hinders waste management optimization" [79] (p. 7). Many of the areas performing landfilling and incineration activities lack adequate technologies. It is difficult to clearly allocate responsibilities on waste management. Articles: [73,75,79–81,85,90,92,93,95,96,98,101]. | Inefficient waste management infrastructures are as well highly relevant for the tourism industry but are not specific in any way to the industry. Gray and black water recycling are also highly relevant for the tourism industry in order to improve the water circularity across the hospitality sector, however the costs are very high compared to the costs of not doing so, making the adoption of this practice unattractive.<br><br>**Negative COVID-19 effect:** More waste has been generated due to the new health and safety measures in place (such as gloves, PPE (personal protective equipment), disposable face masks) and its improper waste management putting more pressure on the environment. For example, one expert mentioned that in certain developing countries that are highly dependent on tourism, the waste infrastructures were not working as they are privately owned and function only with the income generated by the tourism industry. **(Frequency: 3 experts mentioned this COVID-19 effect)**. |
| **Title of CECP:** I.03: (Infrastructure 03)—Difficult to implement CE spatial planning and transportation infrastructure<br>**Area:** Infrastructure<br>**Level of specificity:** Low<br>**Average importance:** 3.50<br>**Average urgency:** 3.33<br>**Frequency of specific CECP:** 6 experts | Lack of spatial planning mechanisms following CE rules and difficulties in managing complex urban systems. "Tension between urban planning and facilitating the kind of experimentation that CE calls for (how to manage the changing economy and the changing structure in a built form)" [81] (p. 2200). The socio-technical lock-in hinders the implementation of CE, "even if there is willingness amongst institutions providing urban infrastructure and services to adopt circular design or integrated approaches, it is practically difficult to alter these infrastructural systems due to the capital cost and disruption generated by such a radical transformation" [80] (p. 10). Articles: [80,81,101]. | Space has been planned without CE in mind, and infrastructures have already been established. Changing this for CE purposes represents an important disruption of the established systems. In addition, important to the tourism industry, but without any tourism specificities. |

Table A3. CECPs on an organizational level.

| Meta Data on CECPs | Explanation of CECP in the Literature | Specificities for the Tourism Industry |
|---|---|---|
| **Title of CECP:** STRAT.01: (Strategy 01)—Shareholder interests not aligned with CE, lack of CE vision<br>**Area:** Strategy<br>**Level of specificity:** High<br>**Average importance:** 3.75<br>**Average urgency:** 3.33<br>**Frequency of specific CECP:** 9 experts | Dealing with a trade-off on whether to have short-term profitability or long-term sustainability. As CE usually involves high short-term costs and low short-term economic benefits instead of low short-term costs and high short-term benefits from a linear economy. "Focus on short-term returns on investment" and "missing the strategic relevance of sustainable development" [106] (p. 1033). CE approaches are not always seen as profitable (e.g., high requirements for pollution reduction and energy saving) and insufficient ROI, which makes it harder to attract investment. This lack of investment power challenges the implementation of CE. In addition, there is a lack of holistic thinking and a multi-stakeholder approach. There is a high focus on individual company interests and a lack of CE vision. Businesses face important amounts of sunk value and sunk cost that have already been invested in suppliers, real capital, and human capital making it very difficult to transform their approach toward CE.<br>Articles: [74,75,80,82,84–86,88,89,93–96,100,101,106,108,112]. | The key players in moving the tourism industry toward CE are hotels. The hotel business is characterized by two main shareholder groups: the owner of the hotels and the shareholders of the brands that operate the hotels. Both shareholders are short-term oriented, for different reasons. Many shareholders are short-term oriented as they seek to optimize their share prize in the short term. Hotel owners on the other side, only have 1–3-year contracts with hotel brands. Thus, they have a great focus to perform well in this time to hopefully renew the contract. Short-term orientation of those two key shareholder groups represents a crucial challenge for moving toward a circular economy in the tourism industry. In addition, experts have mentioned that older generations of hotel owners see the less the need of being sustainable compared to the new generations of hotel owners.<br><br>**Negative COVID-19 effect:** Businesses have changed their priorities with regards to sustainability. They are focusing on survival rather than on making big investments to become a more circular business as they are lacking the financial resources to make the green and digital shift needed to achieve circularity, renovate their buildings, change their portfolio, find other more sustainable suppliers, etc. Moreover, they are also facing a lot of pressure to go back to the status quo in order to maintain the ROI. **(Frequency: 14 experts have mentioned this)**. |

**Table A3.** *Cont.*

| Meta Data on CECPs | Explanation of CECP in the Literature | Specificities for the Tourism Industry |
|---|---|---|
| **Title of CECP:** STRAT.02: (Strategy 02)—Lack of transparency, forecast ability and difficult decision making<br>**Area:** Strategy<br>**Level of specificity:** Low<br>**Average importance:** 3.33<br>**Average urgency:** 3.33<br>**Frequency of specific CECP:** 3 experts | There is a lack of end-to-end visibility and forecast ability hindering the implementation of CE. The challenge of validation not being achievable until further sales make CE adoption riskier. Both poor forecast ability and difficult validation make decision making more challenging to implement CE in the most efficient and effective way. Businesses are not certain that demand or input prices will not go back to past levels and there is uncertain return. "The business cannot be sure what new technologies and business environments will emerge such that if it tries to change too fast now, it will miss a better investment opportunity in the future" [112] (p. 19). Articles: [6,74,107,108,110,112]. | Tourism businesses lack the needed KPIs to measure their circularity. |
| **Title of CECP:** STR.01: (Structure 01)—Missing organizational abilities and structure for CE<br>**Area:** Structure<br>**Level of specificity:** High<br>**Average importance:** 3.33<br>**Average urgency:** 3.33<br>**Frequency of specific CECP:** 3 experts | Depending on how the organization is structured, it will enable or hinder the implementation of CE in their business model. Lack of organizational capabilities that are necessary to implement circular practices across the organization's several functions. "Often, life-cycle practitioners are functionally a part of a company's environment, safety, and health division – separated or disconnected from the process design and product development departments. Thus, the knowledge of the life-cycle practitioners is not shared with developers, and the developers may not be aware of how life-cycle thinking can be integrated into design and development" [109] (p. 20). Articles: [74,93,109]. | The structure of the hotel business in the tourism industry is a key reason why shareholder interests are not aligned with CE initiatives. Managers of hotels are rarely the owners of the hotel (franchise contract); they are rather HR managers. Thus, they do not have the decision power to change the physical setting in which they operate. Furthermore, hotel chains have many levels of hierarchy, making it difficult to get changes approved. If a hotel manager wants to change something, even if it is a small action, he first needs to ask permission from the head office of the chain, and they will evaluate if the decision could have negative effects on the hotel chain's profitability. This slows down the whole decision-making process to take certain circular actions that could benefit the business. |
| **Title of CECP:** CULT.01: (Culture 01)—Lack of managerial commitment and leadership towards CE<br>**Area:** Culture<br>**Level of specificity:** Low<br>**Average importance:** 4.00<br>**Average urgency:** 3.80<br>**Frequency of specific CECP:** 5 experts | From the top down, managerial commitment and weak leadership toward CE are major challenges, which are argued by time constraints and reliance on business leaders to make the CE transition. From the bottom up, heavy organizational hierarchies prevent bottom-up experimenting. "Lack of leadership commitment" [75] (p. 20). Articles: [11,74,75,79,85,87,91,98,106,109,112]. | Lack of managerial commitment is a key issue in the tourism industry. However, the experts did not mention any specific aspect related to tourism. |

**Table A3.** *Cont.*

| Meta Data on CECPs | Explanation of CECP in the Literature | Specificities for the Tourism Industry |
|---|---|---|
| **Title of CECP:** CULT.02: (Culture 02)—Resistance to changes towards CE and conflict with the existing culture<br>**Area:** Culture<br>**Level of specificity:** High<br>**Average importance:** 2.71<br>**Average urgency:** 3.14<br>**Frequency of specific CECP:** 5 experts | Not all organizations are willing to change their business models to make them more or fully circular due to internal resistance to risk among managers and shareholders, rigidity in business routines, and different preferences (preferences for incremental over radical innovation). The aversion to risk is a common resistance challenge toward the adoption of CE due to costly implementations. The hesitant company culture with a predominant linear mindset encourages resistance to change toward CE. The prevailing structures in many industries, known as "linear lock-in", act as a barrier to the implementation of CE. "Conflict of interest within companies" [93] (p. 2). The already settled company culture conflicts with CE adoption due to again risk aversion, and the poor internal cooperation is difficult too. The silo thinking in the business's culture reduces the organization's efficiency. Articles: [6,11,15,80,82,85,87,91,93,97,100,105–107,110]. | The resistance toward change in tourism is deeply rooted in the culture of tourism companies to do everything possible to serve the customer. The direct feedback loops of platforms such as TripAdvisor enforces a culture that fears change. Tourism companies rather have a customer-centric reactive culture than an innovative, proactive one based on their own drivers. |
| **Title of CECP:** CUSTS.01: (Customer Segment 01)—Only niche markets for CE offerings<br>**Area:** Customer Segment<br>**Level of specificity:** Low<br>**Average importance:** 2.00<br>**Average urgency:** 2.50<br>**Frequency of specific CECP:** 2 experts | The majority of customers are not willing to pay higher prices for CE offerings due to three reasons: Segment A is highly price-sensitive with no specific interest in sustainable offerings; Segment B is price-insensitive, caring about the environment but not aware of the impact of their purchase decisions; and Segment C is price-insensitive and aware of its purchase impact but lacks knowledge regarding the environmental impact of different offerings. "Used products are often considered more or less inferior, an idea that is strongly supported by marketing of new products. This preference limits the potential of organizing local collection and exchange of goods" [75] (p. 13) Articles: [12,20,74,75,79,84,87,91,96,97,100,103,104,106,107,110]. | Experts agree that the market demanding circular offerings is also small in the tourism industry, but they do not mention any specificity for the tourism industry. |

Table A3. *Cont.*

| Meta Data on CECPs | Explanation of CECP in the Literature | Specificities for the Tourism Industry |
| --- | --- | --- |
| **Title of CECP:** CUSTS.02: (Customer Segment 02)—Low willingness to pay for CE offerings<br>**Area:** Customer Segment<br>**Level of specificity:** Low<br>**Average importance:** 3.00<br>**Average urgency:** 2.60<br>**Frequency of specific CECP:** 5 experts | Consumers are not always willing to pay a plus for environmentally friendly products as price is a very decisive factor when tacking the final purchase decision. Circular products may be characterized by high selling prices due to enhanced quality (durability) or upgradability, thus constituting a barrier for the customer. It is important to increase the visibility of the benefits of the products to be able to argue for higher prices. "From a demand perspective, a major challenge underlined by the majority of SMEs is the need to create a business case for customers in order to buy a green product or to use a green service[ … ]the need to provide accurate figures and additional evidence of benefits related to green goods and services, the need to convince potential customers that the circular economy approach is the way forward, and the misperception of customers that green products and services are of lower quality than traditional goods and services" [87] (p. 10). Articles: [12,20,79,87,100]. | Experts agree that tourists have a low willingness to pay for CE offerings. One approach would be to lower prices for CE offerings. For instance, experts pointed out that a few nights at a hotel are often more expensive than a monthly rent at home; therefore, tourists expect to have outstanding experiences and are not willing to make compromises to have more CE offerings. |
| **Title of CECP:** CUSTR.01: (Customer Relationships 01)—Difficulty in efficiently branding CE offerings and low status of looped offerings<br>**Area:** Customer Relationships<br>**Level of specificity:** Low<br>**Average importance:** 3.83<br>**Average urgency:** 3.67<br>**Frequency of specific CECP:** 5 experts | CE looped products are not perceived as equally good as new products. In addition, inadequate branding of looped offerings affects the purchasing behavior of consumers. "Low status of products from recycled materials and repaired, reused, refurbished or remanufactured products" [91] (p. 10). Articles: [86,91,100,104,108]. | As the tourism industry relies rather on selling services than products, experts have not mentioned any challenges regarding the perceived status of the quality of the offering. Experts, however, agreed that communicating about CE offerings represents a challenge, as, on the one side, CE is not necessarily a concept that tourists are aware of (is perceived as a confusing buzzword), and on the other side, there is the risk that tourist believes the company is rather interested in greenwashing practices rather than in having a truly circular ambition. |
| **Title of CECP:** VP.01: (Value Proposition 01)—Challenging to create efficient CE value proposition designs for complex offerings<br>**Area:** Value Proposition<br>**Level of specificity:** Low<br>**Average importance:** 3.57<br>**Average urgency:** 3.29<br>**Frequency of specific CECP:** 5 experts | There is limited focus on achieving circularity when it comes to product design. There are many design challenges to durable reuse and recovery products. The product complexity also hinders Life-Cycle Assessments. There is a lack of sufficient guidelines toward product design that enable circularity. "The difficulties related to the use of tools available to support the design of sustainable PSSs (product service systems)" [110] (p. 4) is also a challenge to implement CE. Articles: [15,74,84,86,88,91,99,108,110,111]. | Experts have mentioned that businesses in the tourism sector focus too much on downstream solutions instead of on upstream innovations and on implementing other business models such as "product as a service". |

**Table A3.** *Cont.*

| Meta Data on CECPs | Explanation of CECP in the Literature | Specificities for the Tourism Industry |
|---|---|---|
| **Title of CECP:** VP.02: (Value Proposition 02)—CE offerings often inferior to linear economy offerings<br>**Area:** Value Proposition **Level of specificity:** Low<br>**Average importance:** 3.00<br>**Average urgency:** 3.00<br>**Frequency of specific CECP:** 2 experts | The CE offerings are perceived to have inferior quality, performance, worse customer demand fit, etc. When it comes to redesigning circular products, it is difficult to maintain the same quality level as before. "Worse performance of the services" [102] (p. 924). Articles: [93,99,102]. | It is difficult to maintain a balance between quality, quantity, and sustainability in the tourism sector, which is very dependent on maximizing customer satisfaction and customer loyalty. For instance, it is difficult to provide the quality expected at a food buffet of a hotel chain where customers demand a wide variety of options without generating enormous amounts of food waste. |
| **Title of CECP:** KP.01: (Key Partners 01)—Difficulties in finding suitable CE partners and building solid relationships with them<br>**Area:** Key Partners<br>**Level of specificity:** Low<br>**Average importance:** 4.00<br>**Average urgency:** 3.00<br>**Frequency of specific CECP:** 2 experts | A multi-stakeholder approach is necessary to facilitate the circularity throughout the whole value chain, which has been proven to be very complex when it comes to dealing with the appropriate partners that follow CE principles. Businesses lack the support and long-term cooperation from their key partners. "Companies who decide to move towards CE often experience difficulty in finding appropriate supply chain partners, with appropriate skills and a CE approach" [100] (p. 7404). Articles: [73,79,86,93,100,106,111,112]. | Experts mention that tourism companies seeking to be circular have difficulties finding circular economic suppliers. This is a challenge that is not specific to the tourism industry. |
| **Title of CECP:** KR.01: (Key Resources 01)—Lack of CE related technical resources and know-how<br>**Area:** Key Resources<br>**Level of specificity:** High<br>**Average importance:** 3.62<br>**Average urgency:** 3.38<br>**Frequency of specific CECP:** 6 experts | Companies lack adequate technologies to be able to adopt innovative CE practices. "Need for technical and technological know-how and expertise" [93] (p. 2). Articles: [6,11,12,91,93,94,96–98]. | The discussion around technical resources and know-how in the tourism industry is very much focused on the topic of waste management. Especially the management of food waste from hotels and restaurants plays a prominent role. The key challenge is that tourism companies lack the technologies and know-how to track this waste. Even if some tourism companies invest in waste management technologies, they lack the transparency needed to tell if those technologies have a significant impact. The result is an inefficient use of waste management technologies and low adoption of such technologies in tourism. Furthermore, best practices are not shared among tourism players, keeping the know-how in siloes instead of sharing it. |
| **Title of CECP:** KR.03: (Key Resources 03)—Lack of CE skilled Human Resources within the company<br>**Area:** Key Resources<br>**Level of specificity:** Low<br>**Average importance:** 3.67<br>**Average urgency:** 3.83<br>**Frequency of specific CECP:** 6 experts | Lack of professionals with the necessary skills in CE practices can help the company change its current linear business model toward a circular business model. "Skills shortage to manage the radical innovations needed to transition towards a sustainable, circular economy, for which knowledge often needs to be sourced from outside the organization" [106] (p. 1033). Articles: [87,91,96,99,106]. | No tourism specificity is mentioned by experts. As in other industries, the tourism industry lacks employees trained in a circular economy. |

**Table A3.** *Cont.*

| Meta Data on CECPs | Explanation of CECP in the Literature | Specificities for the Tourism Industry |
|---|---|---|
| **Title of CECP:** RS.01: (Revenue Streams 01)—Cannibalization of sales due to new circular products<br>**Area:** Revenue Streams<br>**Level of specificity:** Low<br>**Average importance:** 1.00<br>**Average urgency:** 1.00<br>**Frequency of specific CECP:** 2 experts | There is a risk of cannibalization due to new circular products diverting sales from existing ones, which can affect the revenue streams obtained from "linear" traditional products, thus reducing the whole future sales of the business. "Risk of cannibalization similar to fashion vulnerability hinders production of long-lasting high-quality products" [107] (p. 5). Articles: [91,98,104,107,108]. | Cannibalization of traditional offerings by circular offerings is also in the tourism industry seen as a challenge. Most importantly, this challenge is seen for travel agencies, who might not be willing to offer circular packages next to their traditional ones. |
| **Title of CECP:** CS.03: (Cost Structure 03)—Very high upfront investment costs to implement CE<br>**Area:** Cost Structure<br>**Level of specificity:** High<br>**Average importance:** 2.83<br>**Average urgency:** 2.50<br>**Frequency of specific CECP:** 6 experts | Very high upfront investment costs hinder the implementation of CE practices, especially in the supply chain. This high upfront investment does not pay back instantly, which blocks investment on CE practices prioritizing investment on linear economy approaches as they usually have short-term economic returns. "High upfront investment costs make 'circular' products more expensive" [88] (p. 4). Recycled materials are generally more expensive in CBM than in linear business models, as acquiring different looped resources and qualified personnel can be more expensive. Furthermore, the lack of capital access to face the high upfront investment costs creates lock-in effect, thus impeding businesses to engage with CE. Articles: [6,15,73,74,82,84,88,93,96–99,103,105,107,108]. | Regarding upfront investments to move toward a circular economy, the tourism industry acknowledges high upfront investments to change assets (such as lower water consumption or higher energy efficiency of hotel buildings). However, experts argue that being a service-oriented industry much can be done by simply changing practices of people working in the tourism industry, which in comparison with asset-heavy product-focused industries require far fewer upfront investments. Especially as a key for moving toward circular economy in tourism seems to be the creation and management of food waste. Experts argue that more circular practices in this field could be simply implemented by training the staff. |

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
