# Peer review of "The 10 Most Crucial Circular Economy Challenge Patterns in Tourism and the Effects of COVID-19"

_sustainability, doi:10.3390/su13094940_

Round 1

Reviewer 1 Report

Dear Authors,

As one of the reviewers I found your submission interesting and read it thoroughly. My general evaluation is positive and I beleive that this manuscript can be published after revisiong some parts of the current version.

My main issues: 1) abstract can be enriched with adding short information about method and findings. 2) the introduction is too long and readers cannot get the central issue. I suggest you to shorten it. 3) theoretical background is not consistent. 2.1 and 2.2 are definitions, while 2.3 is SLR and 2.4 is loosly connected Covid-19. I think you need to restructure this section. I left comments and suggested to extract 2.3 as a new section of literature review, enrich 2.1 and 2,2 with theories and let them as are, and remove 2.4 or connect it theoretically with the research design. 4) methodologicaly, this is no connection between SLR and interviews. normally in systematic grounded theory, SLR used as a means to develop interview questions, but in your case the connection is not observable.

All in all, I beleive that your manuscript can be published after performing requried revision. I would be glad to receive the revised version and review.

Best of luck

I left comments on the attached file. please read and respond them.

Reviewer 2 Report

The reviewer recognized that the manuscript described the several interesting points and discussed the key patterns on CECPs towards transformation of human society on carbon neutrality with view to tourism aspect. Also the reviewer found important discussion by the authors on CECPs in tourism industry against Covid-19 situation which would be key discussion for the current tourism industry at global level.

The reviewer would like to indicate the following comments:

  1. It is vague how and why 33 experts were selected because the manuscript only says, “a good understanding of the tourism industry”. This description does not indicate any scientific- and policy-sound evidence how 33 experts have a good understanding and which level of “a good understanding” they have. The authors need to show its justification further. Also the authors needs to indicate strong justification whether or not number of “33” would be enough for indicating scientific knowledge;
  2. With regard to the matter of waste management, there are 2 sentences and the following discussion which are conflict each other:
    • Ln 566: Increase in waste generation-According to the experts interviewed, three have outlined that COVID-19 has resulted in increasing levels of waste for the tourism industry.
    • Ln 767: In contrast, the bins at touristic locations were nearly not in use due to the lack of tourists. Therefore, shifting waste management assets and adapting systems in a more agile way could help us prevent miss-waste management.
  1. One of the key factors how the human society, including tourism industry, shifts from the liner economy to circular economy, and one of the necessary actions in addition to waste management (downstream management) is recycling (transforming non-value items to value items). However, the manuscript only uses simple word, “inefficient recycling systems”, and there is no any discussion. The reviewer recommends the authors to add discussion on recycling.

Reviewer 3 Report

This is a very interesting and up-to-date paper. Huge effort taken by the Authors to gather sources, carry out interviews, identify Challenge Patterns and point out their practical meaning and usefulness should be highly appreciated. Generally I use quantitative methods in my work, but also appreciate qualitative methods like the ones used here. Especially that grounded theory has been employed to meet qualitative rigors. I believe this is the one of the best papers I had a pleasure to review for an MDPI journal.

Please profread for some minor editorial errors, especially in References: 73, 75, 77 and many more all words in the journal name should begin with capitals. Ref. 30 - lacking number of pages. In some References year is within parantheses (...) in some they are lacking. What is page number in Ref. 53 - 1340003?

Round 2

Reviewer 1 Report

Dear authors,

Thank you for this version. I found all my comments met. You made a good revision. I have no more comments.

Best of luck